# Benchmarking DNA Sequence Models for Causal Variant Prediction in Human Genetics

## Abstract

Machine learning holds immense promise in biology, particularly for the challenging task of identifying causal variants for Mendelian and complex traits. Two primary approaches have emerged for this task: supervised sequence-to-function models trained on functional genomics experimental data and self-supervised DNA language models that learn evolutionary constraints on sequences. However, the field currently lacks consistently curated datasets with accurate labels, especially for non-coding variants, that are necessary to comprehensively benchmark these models and advance the field. In this work, we present TraitGym, a curated dataset of genetic variants that are either known to be causal or are strong candidates across 113 Mendelian and 83 complex traits, along with carefully constructed control variants. We frame the causal variant prediction task as a binary classification problem and benchmark various models, including functional-genomics-supervised models, self-supervised models, models that combine machine learning predictions with curated annotation features, and ensembles of these. Our results provide insights into the capabilities and limitations of different approaches for predicting the functional consequences of genetic variants. We find that alignment-based models CADD and GPN-MSA compare favorably for Mendelian traits and complex disease traits, while functional-genomics-supervised models Enformer and Borzoi perform better for complex non-disease traits. All curated benchmark data, together with training and benchmarking scripts, will be made publicly available upon publication.

## 1 Introduction

Machine learning is increasingly transforming the fields of genomics, human genetics, and healthcare by offering new avenues for predicting the impact of genetic variants on phenotypes and by potentially improving the accuracy of trait or disease risk predictions from individual human genomes. A major challenge in these domains is determining which among millions of intercorrelated genetic variants are causal for Mendelian and complex traits, including diseases. Tackling this challenge, which has profound implications for human health, requires robust and scalable methods that can decode the biological syntax of the human genome and how it drives molecular functions across different cells and tissues.

Three major classes of approaches have been developed to model DNA sequences and predict the effects of genetic variants. The first approach utilizes supervised machine learning models, commonly referred to as sequence-to-function models, which are trained to predict genome-wide functional genomics experimental data from DNA sequences (Eraslan et al., 2019); we refer to these models as *functional-genomics-supervised*. These models predict the functional effects of specific variants by assessing how changes in the DNA sequence influence experimental outcomes. The second approach involves *self-supervised* genomic language models (gLMs), such as masked or autoregressive language models, which are trained only on DNA sequences from one or multiple species without relying on experimental data (Benegas et al., 2024). Models that utilize sequences from multiple species take advantage of evolutionary conservation to gain functional insights. Variant effects in such models are assessed by comparing the log-likelihood between the alternative and reference alleles of the variant, as well as by quantifying changes in the latent representations. Another class of methods includes *integrative* approaches, which combine machine learning predictions with curated annotation features to improve the accuracy of variant effect prediction (Schubach et al., 2024).

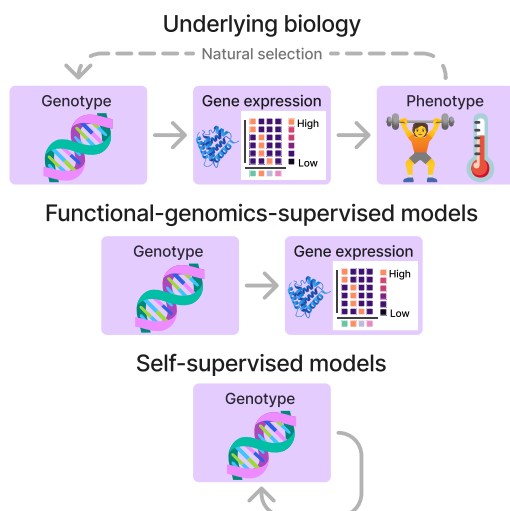

Figure 1: Genotype-to-phenotype relationship and general ML approaches for prediction.

Despite its importance, the field currently lacks consistently processed and comprehensively curated datasets of putative causal genetic variants with reliable labels. Furthermore, there is a pressing need for establishing a common ground for systematically benchmarking state-of-the-art models based on functional-genomics-supervised, self-supervised and integrative approaches, in order to help advance the field.

In this article, we present TraitGym, a curation of two benchmark datasets from human genetics: one comprising causal variants for 113 Mendelian traits, and another consisting of strong causal variant candidates across 83 complex traits, along with carefully constructed control sets matching relevant summary statistics (such as minor allele frequencies, variant types, distances from transcription start sites, and linkage disequilibrium scores) of putative causal variants. We frame the task as binary classification between putatively causal and non-causal variants, allowing to evaluate several state-of-the-art functional-genomics-supervised and self-supervised models, alongside integrative methods and their ensembles. We find that alignment-based integrative and self-supervised models compare favorably for Mendelian traits and complex disease traits, while functional-genomics-supervised models do better on complex non-disease traits. The classification of variants is substantially harder for complex traits, but consistent improvement is observed by ensembling input and predicted features from different models. Additionally, we introduce a new gLM trained specifically on regulatory regions and demonstrate that it compares favorably with other alignment-free self-supervised language models.

## 2 BACKGROUND

One of the essential quests in biology is to understand the genotype-to-phenotype relationship (Figure 1). The genotype is the genetic makeup of an organism, i.e., the set of DNA sequences composing each genome. The phenotype is the collection of observable traits of an individual, such as height or cholesterol levels. Phenotypic variance can be decomposed into components attributed to genetic and environmental factors. The influence of non-coding genetic variants on phenotype is mediated via the expression of genes in different tissues and cell types. Functional-genomics-supervised models attempt to learn the relationship between DNA sequence and gene expression, leveraging genome-wide experimental data (Eraslan et al., 2019). Natural selection closes the loop by impacting which genotypes are favored over time, based on the fitness of the phenotype on a given environment. Therefore, the space of observed DNA sequences contains rich information about the underlying biology; this is precisely the signal leveraged by self-supervised DNA language models (Benegas et al., 2024).

The are two classes of phenotypic traits: Mendelian and complex (Figure 2). Mendelian traits, such as hemophilia, can be strongly affected by a single mutation in a single gene. On the other hand,

complex traits, such as the risk to develop Alzheimer's disease, are affected by several mutations in multiple genes, each typically with a small individual effect. The fact that variants affecting Mendelian traits have larger phenotypic effect sizes than variants affecting complex traits makes the former relatively easier to predict, as they tend to have larger effects on gene expression (the signal picked up by functional-genomics-supervised models) and tend to be subject to stronger purifying selection (the signal picked up by self-supervised models).

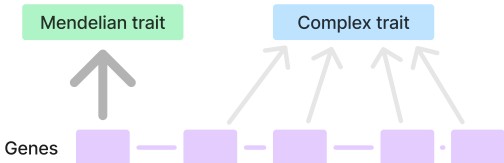

Figure 2: Mendelian vs. complex traits. A single gene typically controls a Mendelian trait, whereas a complex trait is influenced by multiple mutations across several genes, each contributing a small individual effect.

## 3 RELATED WORK

GeneticsGym (Finucane et al., 2024) evaluates the prediction of causal variants for human complex traits, but limited to protein-coding variants. Dey et al. (2020) evaluate the prediction of non-coding causal variants for human complex traits, but limited to a previous generation of functional-genomics-supervised models. Concurrent work (Fabiha et al., 2024) also evaluates the prediction of causal variants for complex traits, but does not cover self-supervised models nor Mendelian traits. Benegas et al. (2023a) evaluate the prediction of non-coding causal variants for human Mendelian traits, but with a much larger, non-subsampled negative set of 2.6 million variants, which makes it less practical to evaluate some of the latest, computationally expensive models.

Tang et al. (2024) benchmark the ability of functional-genomics-supervised and self-supervised models to predict non-coding variant effects on gene expression, but they cover neither Mendelian nor complex traits. BEND (Marin et al., 2024) and GV-Rep (Li et al., 2024) evaluate self-supervised models for the prediction of disease-associated variants from ClinVar (Landrum et al., 2020). While not documented, it is likely that these variants mostly cover Mendelian rather than complex diseases. Furthermore, expert-reviewed pathogenic variants in ClinVar are highly skewed towards coding and splice region variants, containing only a single promoter variant and no intergenic variants (Table A.7). Neither of these benchmarks establishes adequate baselines for this task. BEND includes a single early-generation functional-genomics-supervised model (Zhou & Troyanskaya, 2015), but does not include any conservation-based model, which are usually strong for this task (Benegas et al., 2023a). GV-Rep does not include any baseline.

Thus, TraitGym is the only benchmark of causal non-coding variant prediction for both Mendelian and complex human traits. Furthermore, it is the only available framework to evaluate both the latest functional-genomics-supervised and self-supervised models, as well as strong non-neural baselines.

## 4 BENCHMARK DATASETS

TraitGym consists of two curated datasets of non-coding genetic variants affecting Mendelian and complex traits (Table 1). We focus on non-coding variants since understanding their impact is a particularly important use case for DNA sequence models, compared to coding variants which are more commonly interpreted using protein sequence models. Further, we focus on single-nucleotide variants, the most common form of genetic variation, which is still challenging to interpret. Our data curation process is outlined in Figure 3 and additional details are provided in Appendix A.1.

Table 1: Number of variants and traits in TraitGym.

| Dataset | # putatively causal variants | total # variants | # traits |
|---|---|---|---|
| Mendelian traits | 338 | 3,380 | 113 |
| Complex traits | 1,140 | 11,400 | 83 |

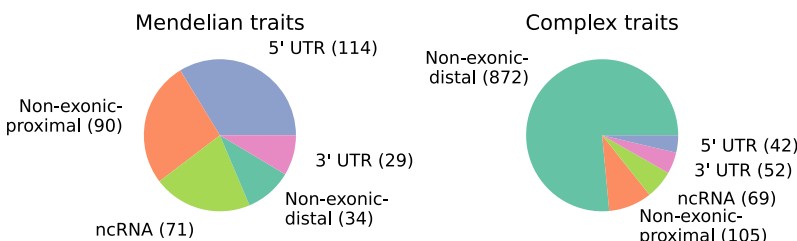

Figure 3: Matching putatively causal and control variants. Nine matched control variants are used for each putatively causal variant, within each chromosome. See the text for the details.

**Mendelian traits**
5' UTR (114)
Non-exonic-proximal (90)
3' UTR (29)
Non-exonic-distal (34)
ncRNA (71)

**Complex traits**
Non-exonic-distal (872)
5' UTR (42)
3' UTR (52)
ncRNA (69)
Non-exonic-proximal (105)

Figure 4: Distribution of consequence classes of putative causal non-coding variants.

**Mendelian traits.** Curated causal non-coding variants for 113 Mendelian diseases were collected from Online Mendelian Inheritance in Man, OMIM (Smedley et al., 2016). For additional stringency, we filtered out a small percentage of variants with minor allele frequency (MAF) greater than 0.1% in the Genome Aggregation Database, gnomAD (Chen et al., 2024). We used gnomAD common variants (MAF > 5%) as controls.

**Complex traits.** Putative causal and control non-coding variants for 83 complex traits were obtained by processing statistical fine-mapping results (Kanai et al., 2021) from association studies in the UK BioBank data (Bycroft et al., 2018). Specifically, we used variants with posterior inclusion probability (PIP) in the credible set greater than 0.9 in any trait as positives and variants with PIP < 0.01 in all traits as controls. We additionally filtered the positive set to genome-wide significant variants ($p < 5 \times 10^{-8}$).

**Variant type (or consequence) annotation.** We annotated the consequence (e.g., intergenic, intronic, 5′ UTR, 3′ UTR, etc.) of each variant using Ensembl (McLaren et al., 2016), and refined this annotation by overlapping with candidate $cis$-regulatory elements from ENCODE (Epstein et al., 2020). Distal non-exonic variants (potential enhancers) comprise a small proportion (10%) in the Mendelian traits dataset but the vast majority (76%) in the complex traits dataset (Figure 4).

**Matching positives and negatives.** For each putative causal non-coding variant, we sampled 9 non-coding variants from the control set, matching chromosome, consequence, and distance to transcription start site (TSS). For complex traits, we additionally matched MAF and linkage disequilibrium (LD) score (Bulik-Sullivan et al., 2015) in the UK BioBank. We sampled only 9 controls per positive variant in order to be able to evaluate even the most computationally demanding models. However, we also provide a larger version of the dataset with millions of negative controls per positive variant, for which we evaluate a subset of the models. This expanded version of the dataset for Mendelian traits does not require any subsampling of negatives, but for complex traits we do subsample to match the MAF distribution (Finucane et al., 2024), while still keeping millions of variants.

**Task definition.** The task is to classify whether a variant is putatively causal for any trait or not. The input data consist of the reference and alternate allele together with the DNA sequence context. As evaluation metric, we calculate the area under the precision recall curve (AUPRC) for each chromosome (for a model trained on the remaining chromosomes), and then compute a weighted average across chromosomes based on sample size, together with a standard error estimated via bootstrapping (described in Appendix A.2.4). The baseline AUPRC is 0.1, which is the proportion of positives.

Table 2: Benchmarked models.

| Model | Dependencies | | | # params | Context size | # extracted features | Source |
|---|---|---|---|---|---|---|---|
| | Functional genomics | Alignment | Population data | | | | |
| **Functional-genomics-supervised models** | | | | | | | |
| Enformer | Yes | No | No | 246M | 196K | 5,138 | Avsec et al. (2021) |
| Sei | Yes | No | No | 890M | 4K | 41 | Chen et al. (2022) |
| Borzoi | Yes | No | No | 186M | 524K | 7,617 | Linder et al. (2023) |
| **Self-supervised models** | | | | | | | |
| GPN-MSA | No | Yes | No | 86M | 128 | 770 | Benegas et al. (2023a) |
| NT | No | No | No | 2.5B | 6K | 2,562 | Dalla-Torre et al. (2023) |
| HyenaDNA | No | No | No | 14M | 160K | 258 | Nguyen et al. (2023) |
| Caduceus | No | No | No | 8M | 131K | 514 | Schiff et al. (2024) |
| gLM-Promoter | No | No | No | 152M | 512 | 1,026 | This work |
| **Integrative models** | | | | | | | |
| CADD | Yes | Yes | Yes | N/A | N/A | 114 | Schubach et al. (2024) |

Table 3: Extracted features and zero-shot scores for each model type.

| Model type | Extracted features | Zero-shot score |
|---|---|---|
| Functional-genomics supervised (Enformer/Borzoi) | $\ell_2$ scores: change in activity in each track $\ell_2$ of $\ell_2$ scores: aggregation of $\ell_2$ scores across several tracks (all + within each assay type) | $\ell_2$ of $\ell_2$ scores (all tracks) |
| Functional-genomics supervised (Sei) | Change in sequence class scores | Max absolute change in sequence class scores |
| Self-supervised | LLR, abs(LLR) Embeddings inner product for each hidden dimension | LLR, abs(LLR) Embeddings inner product, $\ell_2$ distance, cosine distance |
| Integrative | CADD input features, CADD score | CADD score |

## 5 MODELS

We benchmark functional-genomics-supervised models, self-supervised gLMs and integrative models (Table 2). We introduce a new gLM, called gLM-Promoter, trained using the genomes of 434 animal species, following the training objective of GPN (Benegas et al., 2023b) and the ByteNet convolutional architecture (Kalchbrenner et al., 2017; Yang et al., 2024). It is only trained on promoters as an attempt to focus on regulatory regions (we would have liked to train on enhancers as well but no annotation exists for non-model organisms). Additional details on models are provided in Appendix A.2.

We evaluate zero-shot model scores as well as ridge logistic regression classifiers (linear probing) trained using extracted features (Table 3). We use a number of folds equal to the number of chromosomes. In each fold, we test on a single chromosome using a model trained on the remaining chromosomes, and the regularization hyperparameter is chosen based on cross-validation on the training chromosomes (detailed in Appendix A.2.4).

**Functional-genomics-supervised models.** Sequence-to-function models predict activity in thousands of different functional genomic tracks, covering different assays, such as gene expression or chromatin accessibility, in different tissues and cell types. As variant effect prediction features, we calculate the norm (across spatial positions) of the predicted log-fold-change in activity between the reference and the alternate sequence, for each separate track (referred to as "$\ell_2$ score" in Linder et al. (2023)). As zero-shot score, we aggregate the $\ell_2$ scores of different tracks by taking their $\ell_2$ norm ("$\ell_2$ of $\ell_2$ scores"). Sei (Chen et al., 2022) adopts a different variant scoring approach; it maps each sequence into discrete classes, such as promoters or brain-specific enhancers, and scores a variant according to how much it impacts the relative scores of different classes.

**Self-supervised models.** For self-supervised gLMs, a popular zero-shot score is the log-likelihood ratio (LLR) between the alternate and reference allele[1], which has been shown to reflect learned functional constraints, such as transcription factor binding sites (Benegas et al., 2023b). Good results have also been obtained comparing the embeddings of the alternate and reference alleles (Dalla-Torre et al., 2023; Mendoza-Revilla et al., 2024). We evaluate these different scoring approaches for each model (Table A.8) and choose the best performing one when benchmarking against other models (Table A.9). We additionally obtain a high-dimensional featurization of a variant by calculating the inner product (across genomic positions in a given window) between contextual embeddings of the alternate and reference sequences, for each hidden dimension separately.

**Integrative models.** CADD (Schubach et al., 2024) is built on top of a broad set of curated annotations, including conservation, biochemical activity, population-level data as well as predictions from several machine learning models. Utilizing this rich set of input features, CADD is a logistic regression model trained to distinguish proxy-deleterious from proxy-neutral variants. The output of the model is called the CADD score, which we use as zero-shot score. In this paper, we also train our own models using the broad set of CADD *input* features, which we refer to as CADD features even though they are the input, not the output, of CADD.

## 6 RESULTS

**Mendelian traits.** Among zero-shot scores, CADD and GPN-MSA perform the best, but a supervised model trained using CADD input features achieves the best performance when using linear probing (Figure 5). GPN-MSA is a gLM for the human genome that leverages whole-genome sequence alignments across diverse multiple species. Among the models studied in this paper, CADD and GPN-MSA are the only ones explicitly incorporating conservation features, which might be particularly helpful to predict causal variants for Mendelian traits, expected to be under relatively strong purifying selection. Next come the functional-genomics-supervised models Borzoi and Enformer. Alignment-free gLMs come last, with our new gLM-Promoter model clearly performing the best among them. When using a more relaxed MAF cutoff of 1%, only 19 additional positive variants are included, resulting in very similar results (Figure A.1). Also, we explored matching negatives from the same gene rather than from the same chromosome, which required dropping many variants that could not be properly matched, but with similar overall conclusions (Figure A.2).

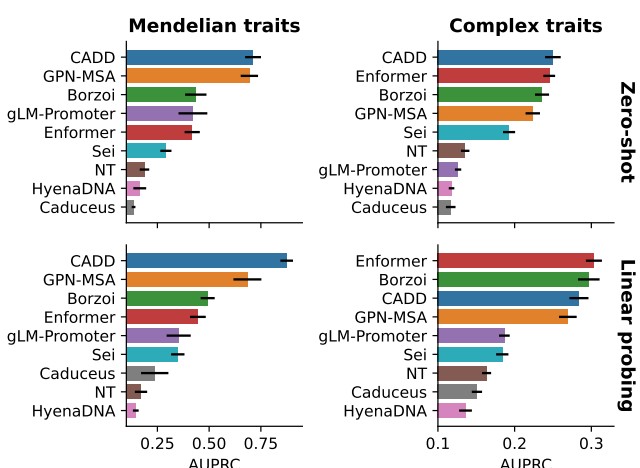

Figure 5: Results on each dataset with zero-shot and linear probing approaches. Zero-shot scores are described in Table 3. For linear probing, we use 113 CADD input features, together with the single CADD output score, while for the other models we only use output features (predicted tracks, LLR or embedding similarity).

CADD is the only model trained on variants and its training variants overlap with around 1% of the variants in our datasets (Table A.10). However, CADD's positives and negatives are not defined based on causal variant annotations (Schubach et al., 2024), and they do not exhibit a clear association with the positive or negative sets in our datasets (Table A.10). We repeated our analysis upon removing this small amount of overlapping variants and found that the aforementioned results remain stable (Figure A.3).

---

[1]The absolute value of the LLR is more appropriate when we want scores to be invariant to which allele is the reference, as in the case of association studies.

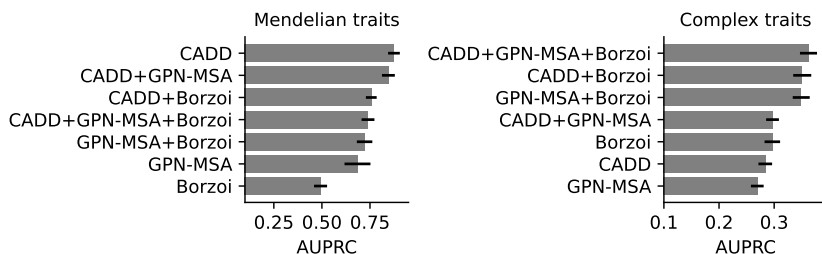

Figure 6: Results of ensembling models by training a logistic regression classifier on the concatenation of their features.

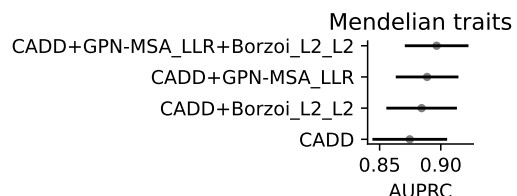

Figure 7: Results of lightweight ensembles. Full CADD features are used, but a much reduced number of features are used from other models.

**Complex traits.** Overall scores are much lower than in Mendelian traits (Figure 5). This is a harder task in principle since variants affecting complex traits are expected to have relatively small effect sizes. CADD and GPN-MSA also perform relatively well on this dataset, but Enformer and Borzoi ultimately come first when their predicted tracks are used in linear probing. Our gLM-Promoter model again does the best among alignment-free gLMs, but only with linear probing. When using a more stringent PIP cutoff, performance generally improves, and Borzoi gains a small relative advantage (Figure A.4). When matching negatives from the same gene rather than the same chromosome, performance is lower overall, but Borzoi obtains a small advantage (Figure A.5).

While the AUPRC is generally recommended for imbalanced datasets where we are mostly interested in the positive minority class (Whalen et al., 2022), we also report the area under the receiver operating characteristic (AUROC). The main difference we see is a slight relative improvement of Enformer and Borzoi zero-shot scores (Figure A.6).

**Results on expanded datasets.** We also considered expanded datasets containing millions of negative controls and evaluated the two models (CADD and GPN-MSA) with precomputed genome-wide zero-shot scores. For Mendelian traits, GPN-MSA achieves a substantial improvement over CADD (Figure A.7). For complex traits, CADD outperforms GPN-MSA, but neither model does very well in absolute terms (Figure A.7). In the future, we hope to evaluate other models on these full datasets, but we estimate that slower models like Caduceus would take approximately 6 months of compute on an NVIDIA A100 80GB GPU.

**Model ensembling.** Given the good performance obtained by different classes of models, potentially leveraging different signals, we evaluated linear probing of combined features extracted from representative models from each class: Borzoi (predicted tracks), GPN-MSA (latent embeddings and LLR) and CADD (input features to the model together with the single output score). The results are summarized in Figure 6. On the complex traits dataset, ensembling the three models achieves the best performance, with a particularly high jump when combining Borzoi with either of CADD or GPN-MSA. On the Mendelian traits dataset, on the other hand, ensembling the full features from different models does not improve upon CADD input features. We attribute this to the fact that (i) the room for improvement is relatively small and (ii) the dataset is small, making it easier to overfit when using high-dimensional features. We refer to the last approach as "full" feature ensembling. However, we do see small improvements when ensembling CADD with a reduced number of features from other models (LLR for GPN-MSA and "$\ell_2$ of $\ell_2$ scores" for Borzoi), which we refer to as "lightweight" feature ensembling (Figure 7).

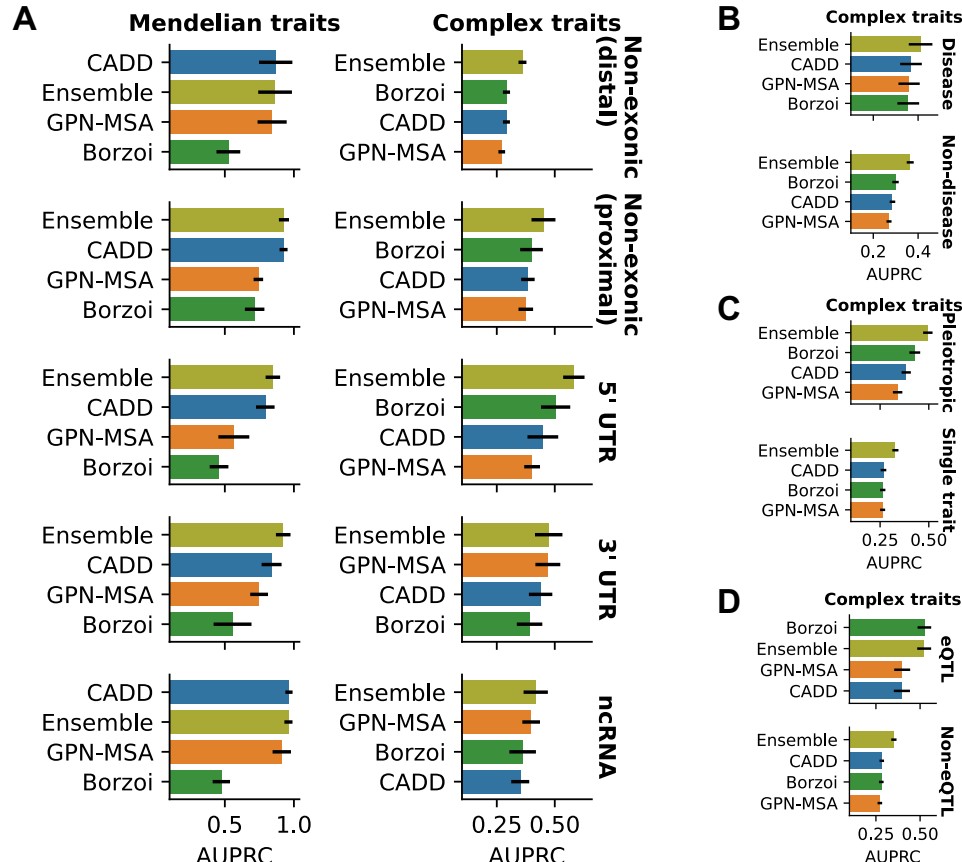

Figure 8: Stratified results. The best score is reported between zero-shot and linear probing. (A) Results by consequence (variant type). Full feature ensemble is evaluated for complex traits, but lightweight feature ensemble is evaluated for Mendelian traits. (B) Results for disease vs. non-disease complex traits. (C) Results for pleiotropic vs. non-pleiotropic variants. (D) Results for complex traits variants stratified by whether or not they overlap with fine-mapped eQTLs.

**Results by consequence (variant type).** We also evaluated the performance stratified by variant consequence classes (Figure 8A). The most important insight here is that the advantage of ensembling for complex traits holds within each consequence class, so it is not simply that different models are experts on different consequences. Second, we note that distal (TSS distance > 1 kb) non-exonic variants for complex traits (which make up the majority) are the hardest class overall. Lastly, while Borzoi performs the worst for Mendelian traits, the gap is the smallest for proximal non-exonic variants.

We also inspected the performance of gLM-Promoter on different consequences, given that it was trained only on promoters (Figure A.8). gLM-Promoter's zero-shot scores perform better on proximal non-exonic and 5' UTR variants, which lie in the regions of the gene with the highest overlap with the model's training data (512 bp around the TSS). Except for the aforementioned classes in Mendelian traits, linear probing outperforms zero-shot scores.

**Results by trait.** We also report performance (Table A.11) for specific traits with sufficiently many putative causal variants and not overlapping too much with each other; specifically, traits with at least 10 causal variants and less than 10% overlap of causal variants with other traits. Ensembling wins in the majority of these traits. Among the 1,140 putative causal variants for complex traits, only 53 affect a disease trait (Table A.1). We evaluated the results stratified by disease vs. non-disease complex traits, pooled given the small sample size (Figure 8B)—for example, our dataset only contains 3 non-coding variants affecting the risk of developing Alzheimer's disease. We note that causal variants for disease traits are easier to classify overall than for non-disease traits, and that

Table 4: Top CADD features in different categories.

| Dataset | Category | Feature | AUPRC | Description |
|---------|----------|---------|-------|-------------|
| Mendelian traits | Alignment | `ZooVerPhyloP` | 0.673 | Conservation in mammals |
| | Functional genomics | `EncodetotalRNA-max` | 0.348 | Max. RNA-seq level |
| | Population data | (-) `Freq100bp` | 0.509 | # common variants within 100bp |
| Complex traits | Alignment | `ZooPriPhyloP` | 0.225 | Conservation in primates |
| | Functional genomics | `EncodeDNase-max` | 0.145 | Max. DNase-seq level |
| | Population data | (-) `Freq10000bp` | 0.131 | # common variants within 10kb |

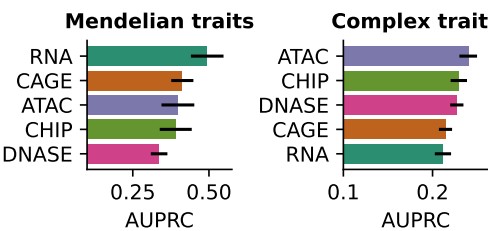

Figure 9: Results of "$\ell_2$ of $\ell_2$ scores" aggregating different assays (Borzoi).

Borzoi loses the edge compared to conservation-aware CADD and GPN-MSA for disease traits. This is consistent with disease traits being under stronger selective pressures. We also noted that putative pleiotropic variants (i.e., those affecting multiple traits) are in general easier to predict, with the biggest advantage being gained by the ensemble model and Borzoi (Figure 8C).

**eQTL colocalization.** We found that 103 putative causal variants for complex traits (9%) overlap with fine-mapped GTEx eQTL variants (Lonsdale et al., 2013; Wang et al., 2021); we found no such overlap for Mendelian trait variants, as expected given their low allele frequencies. The low overlap of complex trait and eQTL variants is well known and Mostafavi et al. (2023) discuss several hypotheses for the cause. We found that eQTL-overlapping variants are much easier to predict than non-eQTL-overlapping variants, across all model types (Figure 8D). We also note that Borzoi achieves a wide margin compared to other models and little is gained from ensembling. We observed that eQTL-overlapping variants are enriched in exonic variants (Fisher's exact $p = 8 \times 10^{-8}$) and, among non-exonic variants, they have lower TSS distances (Mann Whitney $p = 4 \times 10^{-4}$), all of which could explain their increased predictability.

**Interpreting CADD features.** CADD contains informative features from three orthogonal categories: alignment, functional genomics, and population genetic data (Table 4). Conservation features are the most predictive overall. Conservation in mammals is most predictive for Mendelian traits, whereas conservation in primates is most predictive for complex traits. This might be due to the fact that enhancer-like regions, where most causal variants for complex traits lie, tend to only be alignable over shorter evolutionary distances than other functional regions (Phan et al., 2024).

**Interpreting Borzoi features.** We evaluated the performance of aggregated Borzoi scores across specific experimental assays (Figure 9). Of note, gene expression tracks (RNA and CAGE) perform the best on Mendelian traits, while epigenetic tracks (ATAC, CHIP and DNASE) perform the best on complex traits. It has been shown that models such as Borzoi tend to particularly struggle with finding causal variants affecting gene expression when these are distal as opposed to proximal (Karollus et al., 2023). In the case of distal causal variants for complex traits (which make up the majority, see Figure 4), epigenetic tracks might instead be more informative.

A key feature of functional-genomics-supervised models such as Borzoi is that their features are associated with a specific tissue or cell type, which can help interpret disease pathways as well as de-

Table 5: Top three tissue/cell types for different traits, ranked by the highest AUPRC of Borzoi predicted tracks from such tissue/cell type.

| Trait | Tissue/cell type/cell line | AUPRC |
|---|---|---|
| **Mendelian traits** | | |
| Beta-thalassemia | aorta | 0.997 |
| | stomach | 0.988 |
| | adrenal gland | 0.986 |
| Hemophilia B | liver | 1.0 |
| | HepG2 | 1.0 |
| | hepatocyte | 1.0 |
| Hypercholes-terolemia-1 | CD8+ T cell | 0.983 |
| | HepG2 | 0.975 |
| | CD4+ T cell | 0.972 |
| **Complex traits** | | |
| Monocyte count | neutrophil | 0.559 |
| | CD14+ monocyte | 0.559 |
| | HL-60 | 0.559 |
| Hemoglobin A1c | K562 | 0.449 |
| | erythroblast | 0.423 |
| | hematopoietic progenitor | 0.412 |
| High density lipoprotein cholesterol | liver | 0.44 |
| | abdominal adipose tissue | 0.42 |
| | adrenal gland | 0.417 |

sign therapeutics. For traits where Borzoi achieved a good performance, we inspected the tissue/cell type of the top features, and found that they are usually well aligned with previous knowledge (Table 5). For example, the top tissues for high density lipoprotein cholesterol are liver, abdominal adipose tissue and adrenal gland.

## 7 DISCUSSION

**Conclusion.** TraitGym allows to benchmark DNA sequence models on the challenging task of predicting causal variants in human genetics. Alignment-based, conservation-aware models compare favorably on Mendelian traits and complex disease traits, while functional-genomics-supervised models achieve the best performance on complex non-disease traits. A reason for hope in the particularly challenging complex traits dataset is that ensembling predictions and input features from different models yields consistent improvements. We find that alignment-free gLMs are not competitive on causal variant prediction. The best performing model among them—gLM-Promoter, developed in this work—is not the largest gLM, nor does it have a long context. However, one of its defining characteristics is that it was trained only on functional regions; this suggests that, as previously proposed (Tang et al., 2024; Benegas et al., 2024), data curation may warrant more research than architectures. We leave this as promising future work.

**Limitations and future extensions.** The major limitation for benchmarking causal variant prediction for human traits is that the number of known causal variants is small, especially for non-coding regions. In the long term, we expect the number of known causal variants to increase as experimental and statistical techniques improve, together with larger and more diverse patient cohorts. In the short term, we hope to expand the dataset to include variants from other cohorts such as FinnGen (Kurki et al., 2023) and BioBank Japan (Nagai et al., 2017). One of the challenges is that, while many fine-mapping results are publicly available, it is still hard to get access to other quantities such as LD scores, which are important for constructing a rigorous control set.

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

# A  APPENDIX

## A.1  DATASETS

### A.1.1  MENDELIAN TRAITS

Non-coding pathogenic OMIM variants were obtained from Table S6 in Smedley et al. (2016). Common variants were obtained from gnomAD (Chen et al., 2024) (version 3.1.2).

### A.1.2  COMPLEX TRAITS

UK BioBank fine-mapping results (Kanai et al., 2021) were downloaded from https://www. finucanelab.org/data (version: Dec. 3rd, 2019). As recommended to increase fine-mapping accuracy (Kanai et al., 2021), we averaged the posterior inclusion probability (PIP) from FINEMAP (Benner et al., 2016) and SuSiE (Wang et al., 2020), and excluded variants where the two methods disagreed by more than 5%. Complex traits in our dataset that are considered diseases or disorders are shown in Table A.1.

Table A.1: Disease or disorder complex traits in our dataset.

| Trait |
| --- |
| Atrial fibrillation |
| Autoimmune disease (Phecode + Self-reported) |
| Alzheimer disease (LTFH) |
| Asthma |
| Blood clot in the lung |
| Breast cancer |
| Coronary artery disease |
| Colorectal cancer |
| Cholelithiasis |
| Seen doctor (GP) for nerves, anxiety, tension or depression |
| Blood clot in the leg |
| Fibroblastic disorders |
| Glaucoma (Phecode + Self-reported) |
| Hypothyroidism |
| Inflammatory bowel disease |
| Inguinal hernia |
| Insomnia |
| Migraine (Self-reported) |
| Prostate cancer |
| Type 2 diabetes |
| Type 2 diabetes (adjusted by BMI) |

### A.1.3  VARIANT ANNOTATION

Consequences were annotated using Ensembl VEP (McLaren et al., 2016) (release 109.1), using flags --most_severe and --distance 1000 (used to distinguish upstream and downstream from intergenic variants). We only kept non-coding consequences (Table A.2). We discarded splice region variants, such as splice donor variants, as these were very few in number. Coding variants, as well as non-coding variants with a very high expected impact such as in splice donors, are excluded from our analysis.

We refined the annotation of non-exonic variants by checking overlap with each of five different ENCODE candidate *cis*-regulatory element (cCRE) categories (Epstein et al., 2020) (Table A.3). We additionally refined the annotation if a variant overlapped not a cCRE but the 500-bp flank of a cCRE, similar to Finucane et al. (2015). When we match negative controls, we make sure to keep the exact same proportion of consequences, including the distribution of cCRE elements and their

Table A.2: Selected consequences in this study.

| Consequence |
| --- |
| *Non-exonic* |
| `intergenic_variant` |
| `upstream_gene_variant` |
| `downstream_gene_variant` |
| `intron_variant` |
| *Exonic* |
| `5_prime_UTR_variant` |
| `3_prime_UTR_variant` |
| `non_coding_transcript_exon_variant` |

Table A.3: ENCODE cCRE categories.

| Category |
| --- |
| PLS (promoter-like signature) |
| pELS (proximal enhancer-like signature) |
| dELS (distal enhancer-like signature) |
| DNase-H3K4me3 |
| CTCF-only |

flanks. For the analysis of performance by consequence, however, we simplify the categorization of non-exonic variants into proximal (TSS dist. $\leq$ 1 kb) and distal (TSS dist. $>$ 1 kb).

TSS distance was computed with respect to protein coding transcripts only. MAF and LD scores for the UK Biobank computed by the Pan-UK Biobank initiative (Karczewski et al., 2024) were downloaded from `s3://pan-ukb-us-east-1/ld_release/UKBB.EUR.ldscore.ht`.

GTEx fine-mapping results where downloaded from `https://www.finucanelab.org/data`. We used a similar PIP cutoff of 0.9 in any tissue, combined between FINEMAP and SuSiE, to define putative causal eQTL variants.

### A.1.4 MATCHING CONTROLS

Nine negative control variants were sampled for each positive causal variant. Chromosome and consequence were matched exactly. We matched variants with the most similar TSS distance, as well as MAF and LD score in the complex traits dataset. More precisely, we defined a vector space of (TSS distance, MAF, LD score) tuples, applied scikit-learn's robust scaler (Pedregosa et al., 2011), and selected negative variants minimizing the euclidean distance to the positive variant. Table A.4 shows that the matched features have minimal predictive power, as intended. For special cases where there were not enough negative controls to match positive variants for a given chromosome and consequence, we subsampled the positive variants until we had at least nine controls per positive variant.

For the full version of the complex traits dataset, we created 100 equal-size MAF bins and subsampled the negative set until the proportion of variants in each bin was equal to that of the positive set.

### A.2 MODELS

### A.2.1 PUBLISHED MODELS

We downloaded several models from Hugging Face Hub (Wolf et al., 2020) (Table A.5). We downloaded Enformer and Borzoi from gReLU's Model Zoo (Lal et al., 2024). Sei scores were obtained via their web server: `https://hb.flatironinstitute.org/sei`. We obtained CADD

Table A.4: Global AUPRC of matched features, close to baseline (0.1).

| Dataset | Feature | AUPRC |
|---|---|---|
| Mendelian traits | (-) TSS distance | 0.115 |
| Complex traits | (-) TSS distance | 0.104 |
| Complex traits | MAF | 0.101 |
| Complex traits | (-) LD score | 0.104 |

Table A.5: Hugging Face Hub models.

| Model | Hugging Face Hub path |
|---|---|
| GPN-MSA | `songlab/gpn-msa-sapiens` |
| NT | `InstaDeepAI/nucleotide-transformer-2.5b-multi-species` |
| HyenaDNA | `LongSafari/hyenadna-medium-160k-seqlen-hf` |
| Caduceus | `kuleshov-group/caduceus-ps_seqlen-131k_d_model-256_n_layer-16` |

v1.7 scores and annotations from https://krishna.gs.washington.edu/download/CADD/v1.7/GRCh38/whole_genome_SNVs_inclAnno.tsv.gz.

### A.2.2 OUR GLM-PROMOTER MODEL

gLM-Promoter was trained on 512-bp sequences centered at TSSs of protein-coding genes from reference genomes of animal species. TSS coordinates were obtained from the gene annotations available at NCBI Datasets (O'Leary et al., 2024). Species available at NCBI Datasets were subsampled, among those with gene annotations, to keep at most one per family. This resulted in 434 reference genomes. gLM-Promoter's training objective follows GPN: base-pair-level tokenization and masked language modeling of local windows of 512-bp with downweighting of repeat positions (soft-masked in the reference genome). gLM-Promoter's architecture follows ByteNet (Kalchbrenner et al., 2017; Yang et al., 2024), consisting of blocks alternating dilated convolutions and feedforward layers. Hyperparameters are displayed in Table A.6. Training took approximately 2 weeks using 4 NVIDIA A100 40GB GPUs.

Table A.6: gLM-Promoter training hyperparameters

| | |
|---|---|
| Window size | 512 |
| Repeat weight | 0.01 |
| Embedding dimension | 1024 |
| Slim | True |
| Convolutional blocks | 64 |
| Convolutional kernel size (first block) | 9 |
| Convolutional kernel size (remaining blocks) | 5 |
| Convolutional dilation schedule | $1, 2, 4, 8, 16, 32, 64, 128, 1, \ldots$ |
| Optimizer | AdamW |
| Weight decay | 0.01 |
| Batch size | 2048 |
| Steps | 370 K |
| Learning rate | $10^{-3}$ |
| Learning rate warmup | 1 K steps |

### A.2.3 FEATURE EXTRACTION

**Functional-genomics-supervised models.** Let $y_i \in \mathbb{R}_+^L$ be the predicted activity for genomic track $i$ in each of $L$ spatial positions. The "$\ell_2$ score" (Linder et al., 2023) is defined as the norm of the

log-fold-change between the predicted activity for the reference vs. alternate sequences:

$$\ell_2 \text{ score}_i := \left\| \log_2 \left( 1 + \boldsymbol{y}_i^{(\text{alt})} \right) - \log_2 \left( 1 + \boldsymbol{y}_i^{(\text{ref})} \right) \right\| \tag{1}$$

We define the "$\ell_2$ of $\ell_2$ score" as the norm of the $\ell_2$ scores across tracks in a set $\mathbb{A}$ (e.g. all genomic tracks, or all genomic tracks from the same experimental assay):

$$\ell_2 \text{ of } \ell_2 \text{ score}(\mathbb{A}) := \left\| (\ell_2 \text{ score}_i, i \in \mathbb{A}) \right\| \tag{2}$$

For Sei we used the official scores provided in their web server https://hb.flatironinstitute.org/sei.

**Self-supervised models.** We compute the log-likelihood ratio between the reference and alternate alleles:

$$\log \frac{\mathbb{P}(\text{alt})}{\mathbb{P}(\text{ref})} \tag{3}$$

For masked language models, it can be computed from the output probabilities when the variant position is masked. For autoregressive models (HyenaDNA), it can be computed from the likelihood of the entire reference and alternate sequences. We also compute similarity in the embedding space. Let $\boldsymbol{Z} \in \mathbb{R}^{D \times L}$ be the sequence embedding with $D$ hidden dimensions and $L$ spatial positions. For HyenaDNA, an autoregressive model, we take the embedding of the rightmost position (could be interpreted as $L = 1$). We compare the reference and alternate embedding using the Euclidean distance:

$$\left\| \boldsymbol{Z}^{(\text{ref})} - \boldsymbol{Z}^{(\text{alt})} \right\|_F \tag{4}$$

cosine distance:

$$1 - \frac{\langle \boldsymbol{Z}^{(\text{ref})}, \boldsymbol{Z}^{(\text{alt})} \rangle_F}{\left\| \boldsymbol{Z}^{(\text{ref})} \right\|_F \left\| \boldsymbol{Z}^{(\text{alt})} \right\|_F} \tag{5}$$

and innner product:

$$\langle \boldsymbol{Z}^{(\text{ref})}, \boldsymbol{Z}^{(\text{alt})} \rangle_F \tag{6}$$

To obtain a high-dimensional featurization of a variant we calculate the inner product separately for each individual hidden dimension $d$:

$$\langle \boldsymbol{Z}_{d:}^{(\text{ref})}, \boldsymbol{Z}_{d:}^{(\text{alt})} \rangle \tag{7}$$

For both functional-genomics-supervised and self-supervised models, we always average the predictions using the forward vs. reverse strand, to ensure reverse-complement invariance.

### A.2.4 LINEAR PROBING

We train a ridge logistic regression classifier pipeline using scikit-learn (Pedregosa et al., 2011), using default arguments as much as possible (Listing 1). The pipeline starts with imputation (only relevant for CADD input features) and standardization. To choose the regularization hyperparameter, we do a grid search using group K-fold cross-validation, with the groups consisting of the training chromosomes. We use the default number (10)of grid points, but shift the range to allow for heavier regularization given that our regression setting is very high-dimensional.

We repeat the entire pipeline training on all but one chromosome and predicting on the held-out chromosome. At the end we obtain predictions for all chromosomes, but each from a separate logistic regression model. Therefore, instead of calculating a global AUPRC, we calculate the AUPRC within each chromosome, and then perform a weighted average based on sample size. To obtain a standard error, we calculate the standard deviation of the distribution of weighted means performed on 1000 bootstrap samples of chromosomes. To allow easy comparison, we also use the weighted average AUPRC to evaluate zero-shot scores, even though it is not strictly necessary.

We only evaluate zero-shot scores on the full version of the datasets. We obtain standard errors from 100 bootstrap samples within the positive and negative sets, in order to maintain the proportion of positives.

```python
from sklearn.impute import SimpleImputer
from sklearn.linear_model import LogisticRegression
from sklearn.model_selection import GroupKFold, GridSearchCV
from sklearn.pipeline import Pipeline
from sklearn.preprocessing import StandardScaler

def train_logistic_regression(X, y, groups):
    pipeline = Pipeline([
        ('imputer', SimpleImputer(
            missing_values=np.nan, strategy='mean',
            keep_empty_features=True,
        )),
        ('scaler', StandardScaler()),
        ('linear', LogisticRegression(
            class_weight="balanced",
            random_state=42,
        ))
    ])
    Cs = np.logspace(-8, 0, 10)
    param_grid = {
        'linear__C': Cs,
    }
    clf = GridSearchCV(
        pipeline,
        param_grid,
        scoring="average_precision",
        cv=GroupKFold(),
        n_jobs=-1,
    )
    clf.fit(X, y, groups=groups)
    return clf
```

Listing 1: Logistic regression classifier (the default penalty is $\ell_2$).

## A.3 Additional tables and figures

Table A.7: ClinVar "Pathogenic" variant consequences (reviewed by expert panel or practice guideline). ClinVar release: `20240909`.

| consequence | count |
|---|---|
| stop_gained | 1687 |
| missense_variant | 988 |
| splice_donor_variant | 177 |
| splice_acceptor_variant | 157 |
| start_lost | 33 |
| splice_region_variant | 23 |
| splice_donor_5th_base_variant | 22 |
| splice_polypyrimidine_tract_variant | 20 |
| splice_donor_region_variant | 13 |
| intron_variant | 6 |
| synonymous_variant | 5 |
| stop_lost | 3 |
| 3_prime_UTR_variant | 1 |
| upstream_gene_variant | 1 |

Table A.8: AUPRC for different gLM zero-shot scores. In boldface: scores within 1% of best score (for a given model).

| Dataset | Model | LLR | abs(LLR) | L2 dist. | Cosine dist. | Inner prod. |
|---|---|---|---|---|---|---|
| Mendelian traits | GPN-MSA | **0.694** | 0.654 | 0.207 | 0.208 | 0.301 |
| | gLM-Promoter | **0.422** | 0.379 | 0.345 | 0.263 | 0.169 |
| | NT | 0.120 | 0.098 | **0.188** | **0.186** | **0.185** |
| | HyenaDNA | 0.115 | 0.106 | 0.117 | 0.116 | **0.165** |
| | Caduceus | 0.108 | 0.088 | **0.135** | **0.135** | **0.131** |
| Complex traits | GPN-MSA | 0.212 | **0.224** | 0.150 | 0.150 | 0.177 |
| | gLM-Promoter | 0.112 | 0.110 | **0.126** | **0.126** | **0.125** |
| | NT | 0.101 | 0.100 | 0.118 | 0.119 | **0.136** |
| | HyenaDNA | **0.110** | **0.111** | 0.102 | 0.102 | **0.118** |
| | Caduceus | 0.098 | 0.097 | **0.115** | **0.115** | **0.117** |

Table A.9: Selected zero-shot approach for each gLM.

| | Mendelian traits | Complex traits |
|---|---|---|
| GPN-MSA | LLR | abs(LLR) |
| gLM-Promoter | LLR | L2 dist. |
| NT | L2 dist. | Inner prod. |
| HyenaDNA | Inner prod. | Inner prod. |
| Caduceus | L2 dist. | Inner prod. |

Table A.10: Number of overlapping variants with CADD training set.

|  | CADD training positives | CADD training negatives |
|---|---|---|
| Mendelian traits positives | 0 | 0 |
| Mendelian traits negatives | 18 | 19 |
| Complex traits positives | 8 | 1 |
| Complex traits negatives | 79 | 55 |

Table A.11: AUPRC for selected traits (at least 10 causal variants and less than 10% overlap of causal variants with other traits). The best score is reported between zero-shot and linear probing. Full feature ensemble is evaluated for complex traits, but lightweight feature ensemble is evaluated for Mendelian traits. In boldface: scores within 1% of best score.

|  | Borzoi | GPN-MSA | CADD | Ensemble |
|---|---|---|---|---|
| **Mendelian traits** | | | | |
| Hyperferritinemia | 0.315 | 0.965 | **0.981** | **0.985** |
| Beta-thalassemia | 0.927 | 0.796 | 0.926 | **0.955** |
| Pulmonary fibrosis | 0.564 | 0.948 | **1.000** | **1.000** |
| Hemophilia B | 0.914 | 0.709 | **1.000** | 0.991 |
| Cartilage-hair hypoplasia | 0.594 | **0.987** | 0.923 | 0.918 |
| Preaxial polydactyly II | 0.546 | 0.959 | **0.969** | **0.967** |
| Hypercholesterolemia-1 | 0.844 | **0.974** | 0.887 | 0.938 |
| Dwarfism (MOPD1) | 0.484 | **1.000** | **1.000** | **1.000** |
| **Complex traits** | | | | |
| Adult height | 0.292 | 0.383 | **0.407** | 0.339 |
| Platelet count | 0.426 | 0.309 | 0.397 | **0.478** |
| Estimated heel bone mineral density | 0.308 | **0.432** | 0.422 | 0.406 |
| Mean corpuscular volume | 0.434 | 0.319 | 0.391 | **0.454** |
| Monocyte count | **0.561** | 0.404 | 0.375 | 0.535 |
| Hemoglobin A1c | 0.475 | 0.375 | 0.426 | **0.517** |
| Albumin/Globulin ratio | 0.455 | 0.431 | 0.516 | **0.559** |
| High density lipoprotein cholesterol | 0.521 | 0.362 | 0.425 | **0.554** |
| Estimated glomerular filtration rate (cystain C) | 0.457 | 0.456 | 0.421 | **0.470** |
| Alkaline phosphatase | **0.492** | 0.292 | 0.352 | 0.446 |
| Gamma-glutamyl transferase | 0.515 | 0.382 | 0.460 | **0.527** |
| FEV1/FVC ratio | 0.430 | 0.494 | **0.505** | 0.487 |
| Pulse pressure | 0.457 | 0.435 | 0.420 | **0.489** |
| Calcium | **0.468** | 0.433 | 0.425 | 0.408 |
| Albumin | **0.615** | 0.544 | 0.480 | 0.602 |
| Body mass index | 0.344 | **0.514** | 0.436 | 0.499 |
| Balding Type 4 | 0.459 | 0.536 | 0.414 | **0.625** |
| Blood clot in the leg | **0.574** | 0.551 | 0.498 | **0.565** |

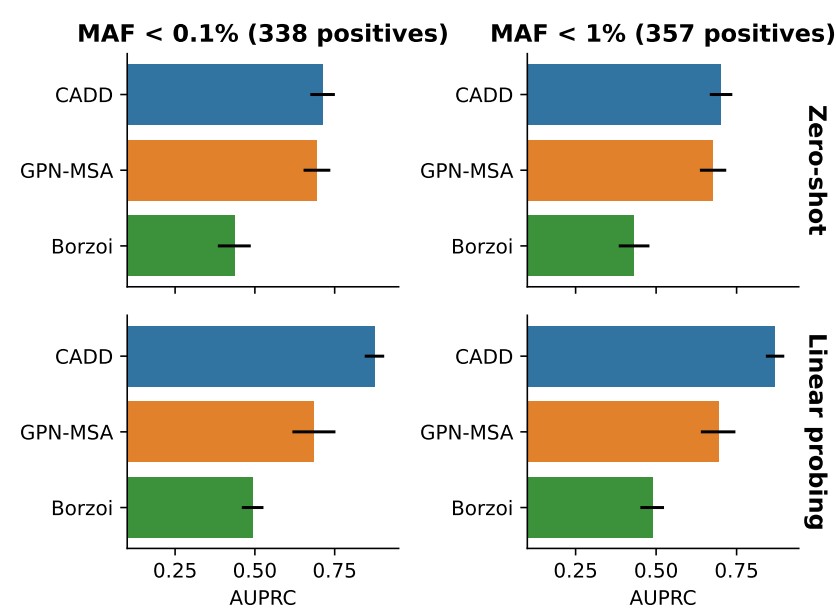

Figure A.1: Ablation of MAF cutoff for positive variants in Mendelian traits dataset.

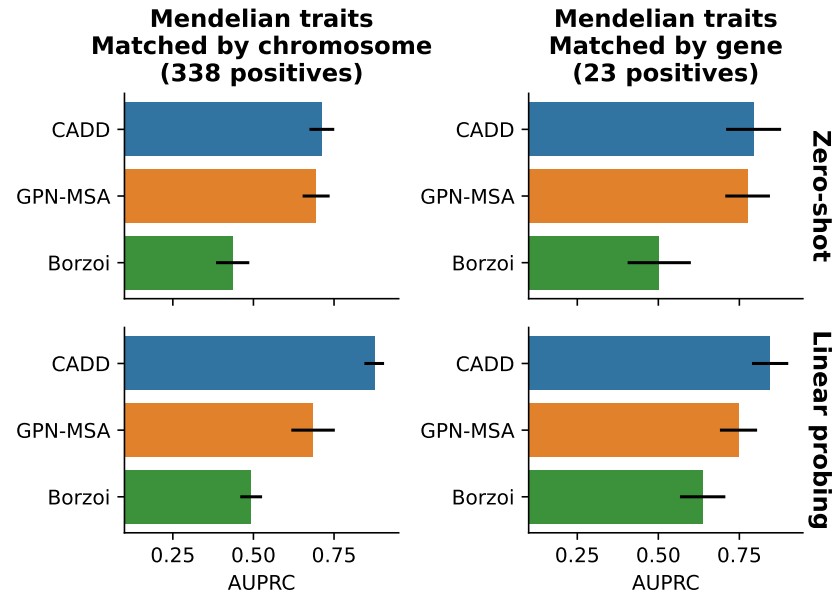

Figure A.2: Mendelian traits results when positive variants are additionally matched by gene (variants that cannot be matched are dropped).

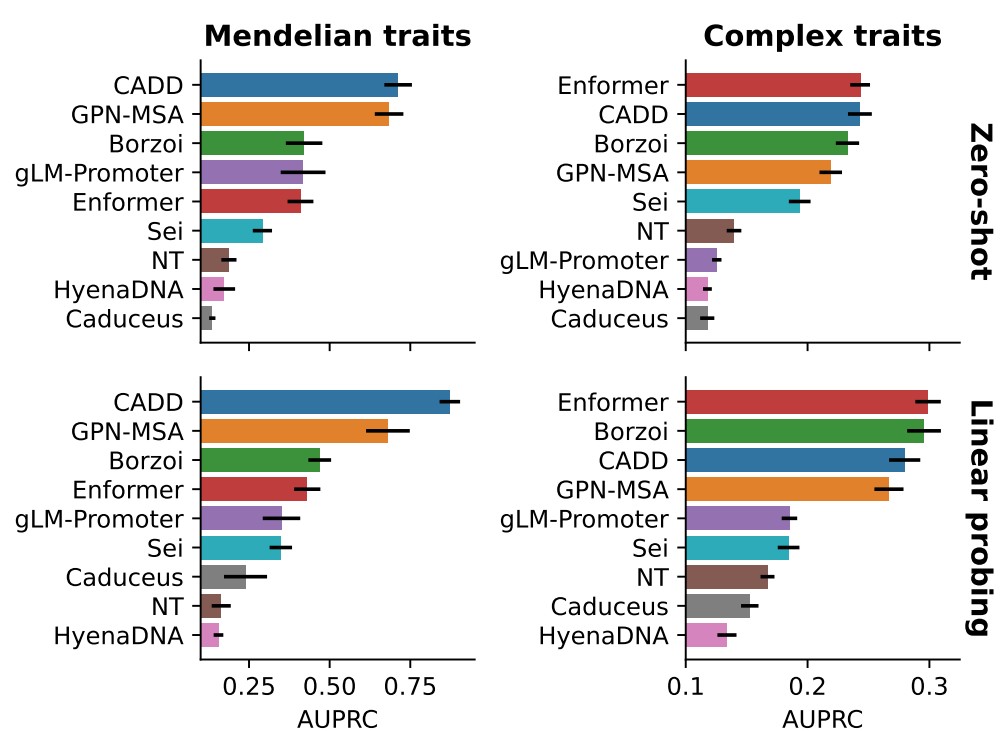

Figure A.3: Results after removing a small amount of variants overlapping CADD training set.

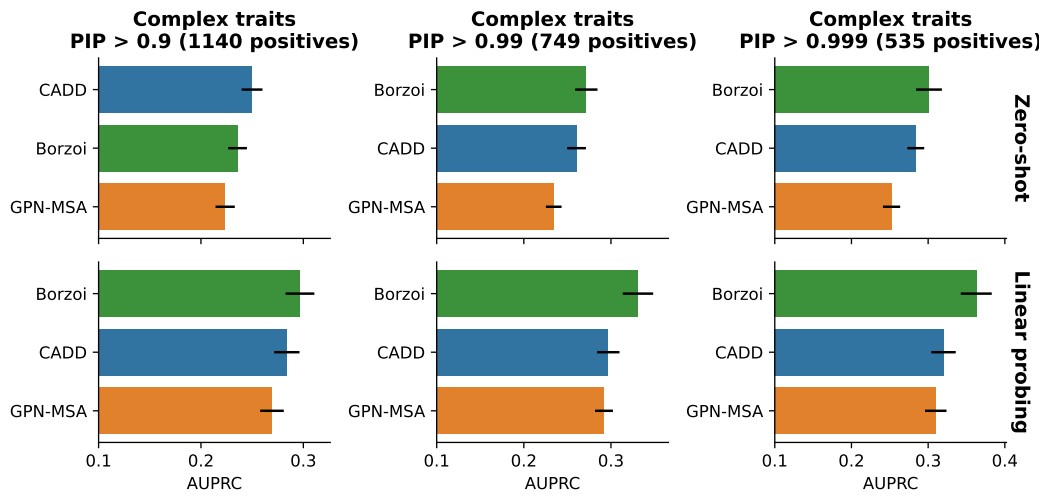

Figure A.4: Results varying the PIP threshold for positive variants.

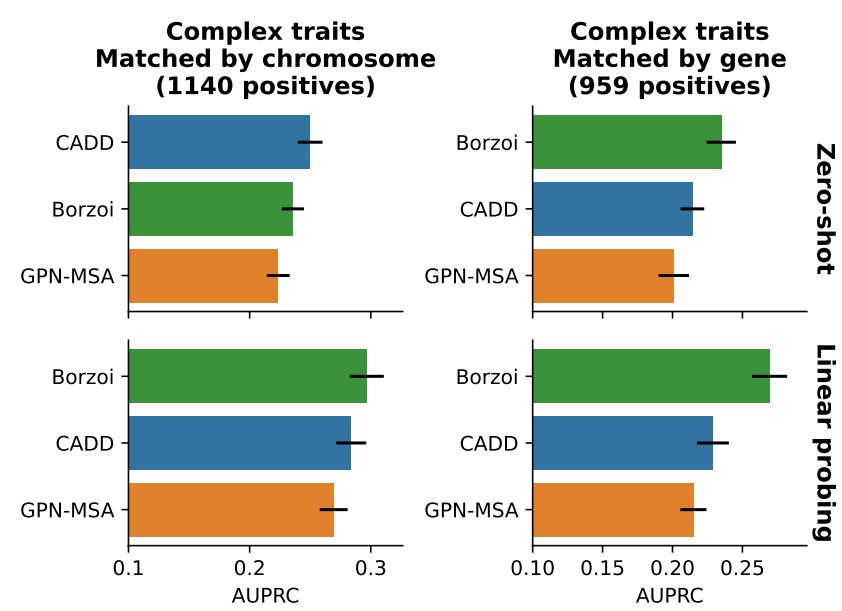

Figure A.5: Complex traits results when positive variants are additionally matched by gene (variants that cannot be matched are dropped).

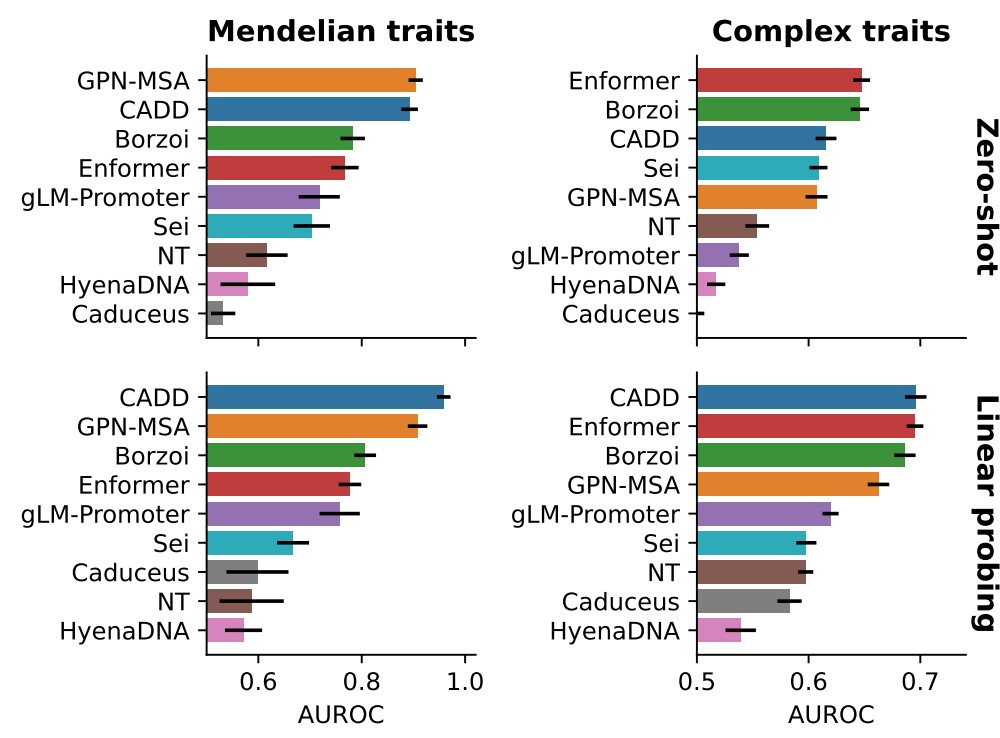

Figure A.6: Results using the AUROC metric.

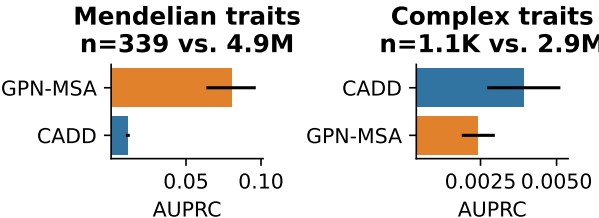

Figure A.7: Results with a much larger negative set of millions of variants. The x-axis range starts at the baseline which is the proportion of positives.

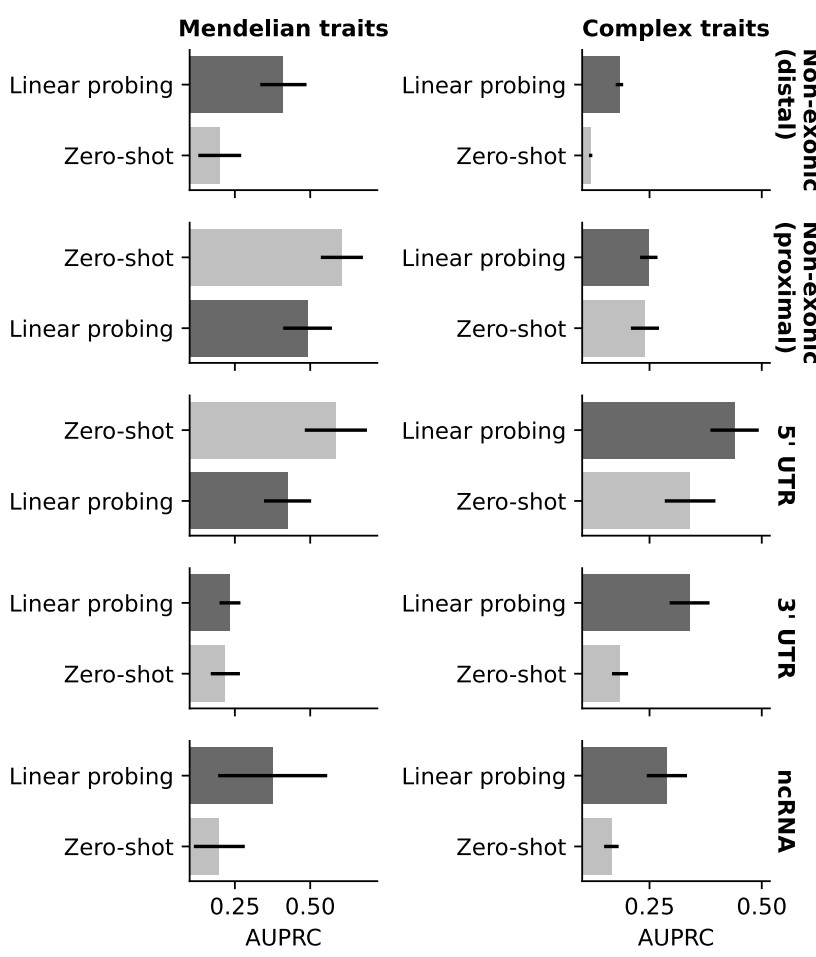

Figure A.8: gLM-Promoter results by consequence.

