# OpenReview forum: "Benchmarking DNA Sequence Models for Causal Variant Prediction in Human Genetics"
_ICLR.cc/2025/Conference — Submitted to ICLR 2025_

### Official Review · Reviewer_3FJZ · 2024-10-27

**Soundness:** 2
**Presentation:** 2
**Contribution:** 2
**Rating:** 3
**Confidence:** 5

**Summary:**

This paper conducted a benchmarking study of DNA sequence models for predicting causal genetic variant for 113 Mendelian and 83 complex traits. They tested the performance of three different types of methods, including functional-genomics-supervised models, self-supervised genomic language models (gLMs), and integrative models that combine machine learning predictions with annotation features (from the viewpoint of ensemble learning). The benchmark results demonstrated differences in model performance across different types of trait, different model classes, and different variant types

**Strengths:**

1.	The consideration of both Mendelian and complex traits provides a quite comprehensive dataset of genetic variants, which is a valuable contribution to the field. Especially the performance difference across different types of traits could deepen the understanding of how a causal genetic variant may contribute to a trait or disease.
2.	The computational problem is well-defined by treating it as a binary classification of causal versus non-causal variants. The negative non-causal variants set should also be very useful in many computational models where negative controls are needed.
3.	The authors demonstrate that combining different types of computational models leads to performance improvements, especially for complex traits. This finding suggests that distinct models capture complementary information that can be leveraged for more accurate predictions.

**Weaknesses:**

1.	The numbers of both variants and traits are small. The way for the benchmark study to define the causal variants is too simple and might be controversial. Both OMIM pathogenic annotation and PIP>0.9 do not imply the variant is causal.
2.	Many baseline methods were missed. The authors may refer to a very recent variant effect prediction benchmark study (doi: 10.1186/s13059-024-03314-7). 24 methods for variant effect prediction were benchmarked. In another benchmark study in 2023 (doi: 10.15252/msb.202211474), 55 variant effect predictors were benchmarked. Many of those methods were ignored in this study.
3.	The evaluation metric used by the benchmark study is too simple based on the binary classification problem. In most of the cases, the number of non-causal variants are far more than causal variants. The auPRC will be affected by the imbalance ratio. Other metrics, such as auROC, precision, recall, f1-score are also necessary to provide more comprehensive benchmark results.
4.	Why only choose 9 non-causal variants as negative controls? The true causal variant proportion for different traits or diseases might be significantly different. Using a constant proportion here for different trait might not be consistent with the true biology.

**Questions:**

1.	The causal variants of Mendelian traits were collected from a single source (Smedley et al., 2016). The causal variants of complex traits were collected from also a single source (UKBiobank). It would be more meaningful if the authors could incorporate more complex traits and disorders with well-defined causal variants, such as those from FinnGen or other large BioBanks, to improve the benchmark’s generalizability.
2.	The authors need to address how biases in the single dataset curation might affect benchmarking results and suggest methods for mitigating these biases, such as incorporating data from additional cohorts or stratified sampling.
3.	Since many deep learning models were included. Any way for enhancing the interpretability of the results by analyzing the biological relevance of model predictions? For instance, interpret how causal variants affect gene regulation networks.

---

> ### Author Response · Authors · 2024-12-02
>
> We appreciate your comments and suggestions.  We have uploaded a revised version of our manuscript.
>
> > The numbers of both variants and traits are small.
>
> > The causal variants of Mendelian traits were collected from a single source (Smedley et al., 2016). The causal variants of complex traits were collected from also a single source (UKBiobank). It would be more meaningful if the authors could incorporate more complex traits and disorders with well-defined causal variants, such as those from FinnGen or other large BioBanks, to improve the benchmark’s generalizability.
>
> > The authors need to address how biases in the single dataset curation might affect benchmarking results and suggest methods for mitigating these biases, such as incorporating data from additional cohorts or stratified sampling.
>
> Thank you for your feedback. We do not believe that the number of traits (196 in total) is considered small in the field. We now include expanded datasets with millions of negative variants, but we note that the number of putative causal variants remains very low. However, we still believe that benchmarking on these putative causal non-coding variants has provided numerous insights throughout the paper. We have done our best to incorporate as many trait-associated variants as possible. We are not aware of any larger publicly available resource of putative causal non-coding variants for Mendelian traits. We also note that the OMIM variants from Smedley et al. (2016) are themselves gathered from the global literature, not from a single source. We envision our data curation and benchmarking as a continued effort, and we hope to add more variants in the future. FinnGen variants, unfortunately, do not have LD scores publicly available yet. The UK BioBank has been a pioneer and contains the largest amount of public statistics available, and was therefore our dataset of choice to kickstart this project.
>
> > The way for the benchmark study to define the causal variants is too simple and might be controversial. Both OMIM pathogenic annotation and PIP>0.9 do not imply the variant is causal.
>
> Thank you for your suggestion. We do clarify that these are putative causal variants:
> “In this work, we present TraitGym, a curated dataset of genetic variants that are either known to be causal or are strong candidates across 113 Mendelian and 83 complex traits, along with carefully constructed control variants.”
> and specifically in the task definition:
> “**Task definition**. The task is to classify whether a variant is putatively causal for any trait or not.”
>
> In some sections, we might speak of causal variants as a shorthand, as this is the norm in the literature. For example, Smedley et al. (2016) also use “causal” in their abstract to refer to the exact same OMIM variants. In the case of complex traits, concurrent work by expert statistical geneticists (Fabiha et al., 2016) also refers to PIP > 0.9 variants as “causal”.
>
> > Many baseline methods were missed. The authors may refer to a very recent variant effect prediction benchmark study (doi: 10.1186/s13059-024-03314-7). 24 methods for variant effect prediction were benchmarked. In another benchmark study in 2023 (doi: 10.15252/msb.202211474), 55 variant effect predictors were benchmarked. Many of those methods were ignored in this study.
>
> We appreciate your suggestion, but we would like to note that the great benchmarks mentioned above concern variant effect prediction for *coding* variants only, while our study is precisely aiming to fill the gap in *non-coding* variant effect prediction.
>
> > The evaluation metric used by the benchmark study is too simple based on the binary classification problem. In most of the cases, the number of non-causal variants are far more than causal variants. The auPRC will be affected by the imbalance ratio. Other metrics, such as auROC, precision, recall, f1-score are also necessary to provide more comprehensive benchmark results.
>
> Thank you for your suggestion. It is correct that in all our datasets the proportion of causal variants of interest is low. A recent review of recommendations for applying ML in genomics recommends the AUPRC in such cases (Whalen et al., 2022). We do not include precision, recall and F1-scores since these require setting specific thresholds while zero-shot scores from different models do not have any obvious cutoffs. We have added AUROC results, however. Please see our updated text:
>
> “While the AUPRC is generally recommended for imbalanced datasets where we are mostly interested in the positive minority class (Whalen et al., 2022), we also report the area under the receiver operating characteristic (AUROC). The main difference we see is a slight relative improvement of Enformer and Borzoi zero-shot scores (Figure A.6).”

---

> > ### Author Response · Authors · 2024-12-02
> >
> > > Why only choose 9 non-causal variants as negative controls? The true causal variant proportion for different traits or diseases might be significantly different. Using a constant proportion here for different trait might not be consistent with the true biology.
> >
> > We appreciate the reviewer’s comment.  Our choice was mainly motivated by computational practicality.  In fact, a recent work from expert statistical geneticists (Fabiha et al., 2024), which appeared while our ICLR submission was under review, also uses a constant number of negatives (specifically, 5 variants) per positive variant. Nevertheless, we now provide expanded negative sets with millions of variants. Please see our expanded section on this topic:
> >
> > “We sampled only 9 controls per positive variant in order to be able to evaluate even the most computationally demanding models. However, we also provide a larger version of the dataset with millions of negative controls per positive variant, for which we evaluate a subset of the models. This expanded version of the dataset for Mendelian traits does not require any subsampling of negatives, but for complex traits we do subsample to match the MAF distribution (Finucane et al., 2024), while still keeping millions of variants.“
> >
> > “**Results on expanded datasets**. We also considered expanded datasets containing millions of negative controls and evaluated the two models (CADD and GPN-MSA) with precomputed genome-wide zero-shot scores. For Mendelian traits, GPN-MSA achieves a substantial improvement over CADD (Figure A.7). For complex traits, CADD outperforms GPN-MSA, but neither model does very well in absolute terms (Figure A.7). In the future, we hope to evaluate other models on these full datasets, but we estimate that slower models like Caduceus would take approximately 6 months of compute on an NVIDIA A100 80GB GPU.”
> >
> > > Since many deep learning models were included. Any way for enhancing the interpretability of the results by analyzing the biological relevance of model predictions? For instance, interpret how causal variants affect gene regulation networks.
> >
> > Thank you for your suggestion. While self-supervised models do not offer any gene-regulatory interpretation, we include results showing that Borzoi can prioritize the tissues most relevant to specific diseases. Please see our text:
> >
> > “A key feature of functional-genomics-supervised models such as Borzoi is that their features are associated with a specific tissue or cell type, which can help interpret disease pathways as well as design therapeutics. For traits where Borzoi achieved a good performance, we inspected the tissue/cell type of the top features, and found that they are usually well aligned with previous knowledge (Table 5). For example, the top tissues for high density lipoprotein cholesterol are liver, abdominal adipose tissue and adrenal gland.”
> >
> > References:
> > Smedley, Damian, et al. "A whole-genome analysis framework for effective identification of pathogenic regulatory variants in Mendelian disease." The American Journal of Human Genetics 99.3 (2016): 595-606.
> > Fabiha, Tabassum, et al. "A consensus variant-to-function score to functionally prioritize variants for disease." bioRxiv (2024): 2024-11.
> > Whalen, Sean, et al. "Navigating the pitfalls of applying machine learning in genomics." Nature Reviews Genetics 23.3 (2022): 169-181.

---

### Official Review · Reviewer_Bb5Y · 2024-10-31

**Soundness:** 2
**Presentation:** 1
**Contribution:** 2
**Rating:** 3
**Confidence:** 4

**Summary:**

TraitGym introduces a benchmarking framework designed to evaluate DNA sequence models in their ability to distinguish causal from control variants across a wide range of traits, covering 113 Mendelian and 83 complex traits. The benchmark relies heavily on established datasets, particularly those curated and described in Finucane et al. (2019, 2024) and databases like gnomAD and OMIM. TraitGym’s primary contributions focus on developing matched negative control sets for variants labeled as pathogenic in previous studies, as well as introducing a new genomic language model, gLM-Promoter. This model claims to specifically train on promoter regions to capture regulatory features that may play a role in causal variant prediction, addressing a gap in current modeling approaches.

The framework evaluates several model types, including functional-genomics-supervised, self-supervised, and integrative models, on a binary classification task aimed at identifying causal versus non-causal variants. To improve predictive performance, TraitGym further explores ensemble approaches, combining features from different models to leverage distinct predictive signals. This work include s an stratification of variant-type and traits to further understand model performance. Additionally, it incorporates a feature interpretation analysis to reveal insights into trait-specific effects across tissues, aiding in the understanding of trait relevance and tissue-specific regulatory mechanisms associated with the variants under study.

**Strengths:**

TraitGym provides a diverse evaluation framework by assessing a range of models, including supervised, self-supervised, and integrative types, enabling a comprehensive comparison across different model architectures. The benchmark’s use of publicly available data sources and well-established resources like gnomAD and OMIM enhances accessibility and reproducibility, contributing to transparency and ease of use for the broader community. The inclusion of the gLM-Promoter model offers a unique approach focused on promoter regions, which may uncover regulatory insights specifically related to these key genomic areas. Additionally, TraitGym’s exploration of ensemble methods, trait specific stratification, and feature interpretation for both CADD and Borzoi is interesting.

**Weaknesses:**

Data Curation and Benchmark Design

The threshold for minor allele frequency (MAF) in pathogenic OMIM variants, set at 0.1%, appears arbitrary, particularly since a 1% threshold is typically standard for rare variants. Conducting ablation studies would clarify the impact of this threshold on model performance and help justify this design choice. The control matching process would also benefit from closest gene-specific stratification. By matching positive and negative samples more closely at the closest gene level, gene-specific biases could be reduced, making the control set construction more rigorous. Finally, the methodology of matching controls based on Euclidean distance across TSS distance, MAF, and LD score assumes these features are equally relevant, which may not hold true. Given that MAF and LD score are more population-dependent, while TSS distance is more functionally related, this approach risks introducing biases in the control set.

Model Evaluation Framework:

The leave-one-chromosome-out (LOCO) validation approach in the linear probing evaluations, while aiming to test model generalization, has notable limitations. Chromosomes differ significantly in size and gene density, leading to inconsistencies in split sizes and class balance for variant groups. Smaller chromosomes, such as chromosome 21, have fewer variants, which can skew model performance and impact the validity of linear probing. Furthermore, features like minor allele frequency (MAF) and linkage disequilibrium (LD) score vary by chromosome and could add further bias if not carefully accounted for during these experiments.

Additionally, the evaluation of the CADD model introduces a potential risk of data leakage. CADD is a meta-predictor that integrates conservation scores and population data, among other annotations, and if any benchmark variants overlap with those in CADD’s training data, the evaluation may be inadvertently biased in CADD's favor. A thorough check of these overlaps would mitigate this risk and ensure an unbiased comparison.

gLM-Promoter Model Clarity

The gLM-Promoter model’s design and training process lack sufficient detail, which weakens its clarity and potential impact. Missing information includes the model's approach to defining promoter regions, the rationale behind choosing a 512bp window size, and how TSS coordinates are determined across different genomes. The current evaluation appears limited and does not specify if the model was exclusively tested on promoter variants, nor does it present a comparison across different variant types. Furthermore, it is unclear whether the evaluated variants are distinct from those seen in the model’s training, particularly when using the human reference genome. Without a clearer contribution to the benchmark, the inclusion of gLM-Promoter could be reconsidered unless region-specific analysis is expanded.

Organization and Clarity Issues

Several presentation issues detract from the paper's readability and clarity. Figure 1, for instance, is overly complex and could benefit from simplification to improve interpretability. Additionally, the writing style is often verbose, and adopting a more concise approach would enhance clarity. For example, the parenthetical explanation of Mendelian versus complex traits in the introduction disrupts the flow unnecessarily and could be integrated more smoothly into the text.

**Questions:**

Could the authors clarify whether variants used in the evaluation overlap with those in CADD’s training data? This would help ensure unbiased results by addressing potential data leakage.

Why was a threshold of nine negative controls chosen per causal variant? An explanation of how this number impacts benchmark robustness would provide greater clarity on the control selection process.

Were ablation studies conducted to assess the impact of the 0.1% MAF threshold for OMIM variants on model performance? A comparison with a 1% threshold would align with conventional standards and strengthen the study’s design choices.

Were any ablation studies conducted to explore a range of posterior inclusion probability (PIP) thresholds for the complex trait benchmark? Examining how different PIP thresholds impact model performance could provide valuable insights into the robustness of the benchmark design for complex traits.

Could the authors provide additional information on their approach for trait- and tissue-based evaluations? Including positive and negative splits for these analyses, as well as integrating plausibly causal eQTLs (e.g., from GTEx fine-mapping), could enhance interpretability and contribute novel insights into trait- and tissue-specific variant effects.

Would the authors consider further expanding the trait analysis by integrating eQTL datasets (such as GTEx fine-mapping) that overlap with causal variants from Finucane et al. (2015) or OMIM? This approach could create a unique dataset combining eQTLs with fine-mapped UKBB GWAS variants, potentially providing additional validation and highlighting cases where expression changes may not fully capture phenotypic effects at the organism level.

---

> ### Author Response · Authors · 2024-12-02
>
> Thank you for your thorough review and insightful suggestions.  We have revised our manuscript and uploaded it above.
>
> > The threshold for minor allele frequency (MAF) in pathogenic OMIM variants, set at 0.1%, appears arbitrary, particularly since a 1% threshold is typically standard for rare variants. Conducting ablation studies would clarify the impact of this threshold on model performance and help justify this design choice.
>
> > Were ablation studies conducted to assess the impact of the 0.1% MAF threshold for OMIM variants on model performance? A comparison with a 1% threshold would align with conventional standards and strengthen the study’s design choices.
>
> Thank you for your suggestion. We have added the suggested ablation, which yields very similar results. We note that the 0.1% cutoff is often used in the literature, e.g. in a rare disease screening application (Adhikari et al., 2020). Please see our updated text below:
>
> “When using a more relaxed MAF cutoff of 1%, only 19 additional positive variants are included, resulting in very similar results (Figure A.1)“
>
> > The control matching process would also benefit from closest gene-specific stratification. By matching positive and negative samples more closely at the closest gene level, gene-specific biases could be reduced, making the control set construction more rigorous.
>
> Thank you for your suggestion. We have included additional versions of the dataset matched by gene. This analysis is certainly of biological interest but requires dropping many causal variants that cannot be properly matched, especially for Mendelian traits. Please see our updated text:
>
> “Also, we explored matching negatives from the same gene rather than from the same
> chromosome, which required dropping many variants that could not be properly matched, but with similar overall conclusions (Figure A.2).“
>
> “When matching negatives from the same gene rather than the same chromosome, performance is lower overall, but Borzoi obtains a small advantage (Figure A.5).”
>
> > Finally, the methodology of matching controls based on Euclidean distance across TSS distance, MAF, and LD score assumes these features are equally relevant, which may not hold true. Given that MAF and LD score are more population-dependent, while TSS distance is more functionally related, this approach risks introducing biases in the control set.
>
> Thank you for your suggestion. We acknowledge that matching multiple features always involves a tradeoff. We empirically find that our approach, which includes robust scaling, results in the matched features (TSS distance, MAF, LD score) themselves having very low predictive power (Table A.4), as intended.
>
> > The leave-one-chromosome-out (LOCO) validation approach in the linear probing evaluations, while aiming to test model generalization, has notable limitations. Chromosomes differ significantly in size and gene density, leading to inconsistencies in split sizes and class balance for variant groups. Smaller chromosomes, such as chromosome 21, have fewer variants, which can skew model performance and impact the validity of linear probing. Furthermore, features like minor allele frequency (MAF) and linkage disequilibrium (LD) score vary by chromosome and could add further bias if not carefully accounted for during these experiments.
>
> Class proportion is maintained since we match each positive with a fixed number of negatives from the same chromosome. To account for differences in sample sizes between different chromosomes, we perform a weighted average of AUPRC. MAF and LD score naturally vary, but there is no reason to believe that they would vary in a biased manner towards a specific chromosome (fine-mapping results already exclude sex chromosomes and the MHC locus). We also note that a very similar LOCO approach has been adopted by expert statistical geneticists in a recent preprint (Fabiha et al., 2024), which appeared while our ICLR submission was under review.

---

> > ### Author Response · Authors · 2024-12-02
> >
> > > Additionally, the evaluation of the CADD model introduces a potential risk of data leakage. CADD is a meta-predictor that integrates conservation scores and population data, among other annotations, and if any benchmark variants overlap with those in CADD’s training data, the evaluation may be inadvertently biased in CADD's favor. A thorough check of these overlaps would mitigate this risk and ensure an unbiased comparison.
> >
> > > Could the authors clarify whether variants used in the evaluation overlap with those in CADD’s training data? This would help ensure unbiased results by addressing potential data leakage.
> >
> > Thank you for your suggestion. Please see our updated text:
> >
> > “CADD is the only model trained on variants and its training variants overlap with around 1% of
> > the variants in our datasets (Table A.10). However, CADD’s positives and negatives are not defined based on causal variant annotations (Schubach et al., 2024), and they do not exhibit a clear association with the positive or negative sets in our datasets (Table A.10). We repeated our analysis upon removing this small amount of overlapping variants and found that the aforementioned results remain stable (Figure A.3).”
> >
> > > The gLM-Promoter model’s design and training process lack sufficient detail, which weakens its clarity and potential impact. Missing information includes the model's approach to defining promoter regions, the rationale behind choosing a 512bp window size, and how TSS coordinates are determined across different genomes. The current evaluation appears limited and does not specify if the model was exclusively tested on promoter variants, nor does it present a comparison across different variant types. Furthermore, it is unclear whether the evaluated variants are distinct from those seen in the model’s training, particularly when using the human reference genome. Without a clearer contribution to the benchmark, the inclusion of gLM-Promoter could be reconsidered unless region-specific analysis is expanded.
> >
> > Thank you for your suggestions. We have expanded our description of the model:
> >
> > “gLM-Promoter was trained on 512-bp sequences centered at TSSs of protein-coding genes from reference genomes of animal species. TSS coordinates were obtained from the gene annotations available at NCBI Datasets (O’Leary et al., 2024). Species available at NCBI Datasets were subsampled, among those with gene annotations, to keep at most one per family. This resulted in 434 reference genomes. gLM-Promoter’s training objective follows GPN: base-pair-level tokenization and masked language modeling of local windows of 512-bp with downweighting of repeat positions (soft-masked in the reference genome). gLM-Promoter’s architecture follows ByteNet (Kalchbrenner et al., 2017; Yang et al., 2024), consisting of blocks alternating dilated convolutions and feedforward layers. Hyperparameters are displayed in Table A.6. Training took approximately 2 weeks using 4 NVIDIA A100 40GB GPUs.”
> >
> > We also analyzed its performance across variant consequences:
> >
> > “We also inspected the performance of gLM-Promoter on different consequences, given that it was trained only on promoters (Figure A.8). gLM-Promoter’s zero-shot scores perform better on proximal non-exonic and 5’ UTR variants, which lie in the regions of the gene with the highest overlap with the model’s training data (512 bp around the TSS). Except for the aforementioned classes in Mendelian traits, linear probing outperforms zero-shot scores.”
> >
> > We also want to clarify that the model was only trained on reference genomes without any human genetic variation data.
> >
> > > Several presentation issues detract from the paper's readability and clarity. Figure 1, for instance, is overly complex and could benefit from simplification to improve interpretability. Additionally, the writing style is often verbose, and adopting a more concise approach would enhance clarity. For example, the parenthetical explanation of Mendelian versus complex traits in the introduction disrupts the flow unnecessarily and could be integrated more smoothly into the text.
> >
> > Thank you for your feedback. We have removed the parenthetical explanation in the introduction, although we still believe that Figure 1 contains a reasonable amount of information.

---

> > > ### Author Response · Authors · 2024-12-02
> > >
> > > > Why was a threshold of nine negative controls chosen per causal variant? An explanation of how this number impacts benchmark robustness would provide greater clarity on the control selection process.
> > >
> > >
> > > We appreciate the reviewer’s comment.  Our choice was mainly motivated by computational practicality.  In fact, a recent work from expert statistical geneticists (Fabiha et al., 2024), which appeared while our ICLR submission was under review, also uses a constant number of negatives (specifically, 5 variants) per positive variant. Nevertheless, we now provide expanded negative sets with millions of variants. Please see our expanded section on this topic:
> > >
> > >
> > > “We sampled only 9 controls per positive variant in order to be able to evaluate even the most computationally demanding models. However, we also provide a larger version of the dataset with millions of negative controls per positive variant, for which we evaluate a subset of the models. This expanded version of the dataset for Mendelian traits does not require any subsampling of negatives, but for complex traits we do subsample to match the MAF distribution (Finucane et al., 2024), while still keeping millions of variants.“
> > >
> > > “**Results on expanded datasets**. We also considered expanded datasets containing millions of negative controls and evaluated the two models (CADD and GPN-MSA) with precomputed genome-wide zero-shot scores. For Mendelian traits, GPN-MSA achieves a substantial improvement over CADD (Figure A.7). For complex traits, CADD outperforms GPN-MSA, but neither model does very well in absolute terms (Figure A.7). In the future, we hope to evaluate other models on these full datasets, but we estimate that slower models like Caduceus would take approximately 6 months of compute on an NVIDIA A100 80GB GPU.”
> > >
> > > > Were any ablation studies conducted to explore a range of posterior inclusion probability (PIP) thresholds for the complex trait benchmark? Examining how different PIP thresholds impact model performance could provide valuable insights into the robustness of the benchmark design for complex traits.
> > >
> > > Thank you for your suggestion. PIP > 0.9 vs. PIP < 0.01 is a standard threshold in the field, used for eQTLs in Enformer (Avsec et al., 2021) as well as for complex traits in two recent papers by expert statistical geneticists (Siraj et al., 2023; Fabiha et al., 2024). We now include additional analyses for more stringent PIP cutoffs. Please see our updated text:
> > >
> > > “When using a more stringent PIP cutoff, performance generally improves, and Borzoi gains a small relative advantage (Figure A.4).”
> > >
> > > > Could the authors provide additional information on their approach for trait- and tissue-based evaluations? Including positive and negative splits for these analyses, as well as integrating plausibly causal eQTLs (e.g., from GTEx fine-mapping), could enhance interpretability and contribute novel insights into trait- and tissue-specific variant effects.
> > >
> > > > Would the authors consider further expanding the trait analysis by integrating eQTL datasets (such as GTEx fine-mapping) that overlap with causal variants from Finucane et al. (2015) or OMIM? This approach could create a unique dataset combining eQTLs with fine-mapped UKBB GWAS variants, potentially providing additional validation and highlighting cases where expression changes may not fully capture phenotypic effects at the organism level.
> > >
> > > Thank you for your suggestions. With regards to the trait-based analysis, for each trait we compare the set of variants associated with the trait, to their matched negatives. The analysis of tissues prioritized by Borzoi is done with respect to specific traits as just described. We now include a section on GTEx eQTL colocalization. Please see our updated text:
> > >
> > > “**eQTL colocalization**. We found that 103 putative causal variants for complex traits (9%) overlap with fine-mapped GTEx eQTL variants (Lonsdale et al., 2013; Wang et al., 2021); we found no such overlap for Mendelian trait variants, as expected given their low allele frequencies. The low overlap of complex trait and eQTL variants is well known and Mostafavi et al. (2023) discuss several hypotheses for the cause. We found that eQTL-overlapping variants are much easier to predict than non-eQTL-overlapping variants, across all model types (Figure 8D). We also note that Borzoi achieves a wide margin compared to other models and little is gained from ensembling. We observed that eQTL-overlapping variants are enriched in exonic variants (Fisher’s exact p = 8 × 10^−8) and, among non-exonic variants, they have lower TSS distances (Mann Whitney p = 4 × 10^−4), all of which could explain their increased predictability.”

---

> > > > ### Author Response · Authors · 2024-12-02
> > > >
> > > > References:
> > > > Adhikari, Aashish N., et al. "The role of exome sequencing in newborn screening for inborn errors of metabolism." Nature Medicine 26.9 (2020): 1392-1397.
> > > > Avsec, Žiga, et al. "Effective gene expression prediction from sequence by integrating long-range interactions." Nature Methods 18.10 (2021): 1196-1203.
> > > > Siraj, Layla, et al. "Functional dissection of complex and molecular trait variants at single nucleotide resolution." bioRxiv (2023).
> > > > Fabiha, Tabassum, et al. "A consensus variant-to-function score to functionally prioritize variants for disease." bioRxiv (2024): 2024-11.

---

### Official Review · Reviewer_Ub7o · 2024-11-04

**Soundness:** 3
**Presentation:** 3
**Contribution:** 3
**Rating:** 6
**Confidence:** 4

**Summary:**

This paper describes the curation of two benchmark datasets for evaluating approaches to predicting causal genetic variants. A study was performed to compare several such existing methods (including one trained by themselves) from three categories using the proposed benchmarks.

**Strengths:**

-  After made available, the two curated benchmark datasets can help to advance the field of developing computational approaches to identifying genetic variants with major functional consequences.

- It is interesting to see that considering alignment is important in self-supervised training for causal variants prediction. (By comparing gLM-Promoter and GPN-MSA if I understand correctly.)

- Analyzed model predictions from varying prospectives and the obtained results are interesting

- It is interesting to see the ensembling of outputs of different models leads to the best predictions.

**Weaknesses:**

- The technical contribution may be considered low with minimal innovation.

- More discussion about gLM promoter and its difference from GPN-MSA and other self-supervised trained models would be helpful.

- A cross comparison between the curated genetic variants with those from ClinVar would strengthen this study.

**Questions:**

- Why choosing to use common variants as controls in the study of Mendelian traits? The causal variants of Mendelian traits are typically rare (i.e., low MAF).

---

> ### Author Response · Authors · 2024-12-02
>
> We appreciate your thoughtful review and suggestions.  We have revised our manuscript and uploaded it above.
>
> > The technical contribution may be considered low with minimal innovation.
>
> We would like to emphasize that our main contribution is the much-needed systematic benchmarking of both the latest functional-genomics-supervised and self-supervised models on the task of predicting causal variants in human genetics.  Our results provide insights into the capabilities and limitations of different approaches, and we believe that this kind of benchmarking study will facilitate future technical work that will help to advance the field.
>
> > More discussion about gLM promoter and its difference from GPN-MSA and other self-supervised trained models would be helpful.
>
> We have now expanded the description of the model. Please see our updated text:
>
> “gLM-Promoter was trained on 512-bp sequences centered at TSSs of protein-coding genes from reference genomes of animal species. TSS coordinates were obtained from the gene annotations available at NCBI Datasets (O’Leary et al., 2024). Species available at NCBI Datasets were subsampled, among those with gene annotations, to keep at most one per family. This resulted in 434 reference genomes. gLM-Promoter’s training objective follows GPN: base-pair-level tokenization and masked language modeling of local windows of 512-bp with downweighting of repeat positions (soft-masked in the reference genome). gLM-Promoter’s architecture follows ByteNet (Kalchbrenner et al., 2017; Yang et al., 2024), consisting of blocks alternating dilated convolutions and feedforward layers. Hyperparameters are displayed in Table A.6. Training took approximately 2 weeks using 4 NVIDIA A100 40GB GPUs.”
>
> We now also analyze its performance across variant consequences. Please see our updated text:
>
> “We also inspected the performance of gLM-Promoter on different consequences, given that it was trained only on promoters (Figure A.8). gLM-Promoter’s zero-shot scores perform better on proximal non-exonic and 5’ UTR variants, which lie in the regions of the gene with the highest overlap with the model’s training data (512 bp around the TSS). Except for the aforementioned classes in Mendelian traits, linear probing outperforms zero-shot scores.”
>
> > A cross comparison between the curated genetic variants with those from ClinVar would strengthen this study.
>
> We now include an investigation into expert-curated ClinVar variants, finding that they lack coverage of the regulatory genome. Please see our updated text:
>
> “Furthermore, expert-reviewed pathogenic variants in ClinVar are highly skewed towards coding and splice region variants, containing only a single promoter variant and no intergenic variants (Table A.7).”
>
> > Why choosing to use common variants as controls in the study of Mendelian traits? The causal variants of Mendelian traits are typically rare (i.e., low MAF).
>
> That is correct. The reason why we use MAF > 5% variants as controls is that this is a recommended standalone criterion (BA1) to classify a variant as benign for Mendelian disorders [1]. This negative set is common in the literature, see e.g. CADD [2]. It would indeed be interesting to use *benign* rare variants as controls, but, in practice, we are not aware of any robust standard approach to confidently call rare *benign* variants.
>
> References:
> [1] Richards, Sue, et al. "Standards and guidelines for the interpretation of sequence variants: a joint consensus recommendation of the American College of Medical Genetics and Genomics and the Association for Molecular Pathology." Genetics in Medicine 17.5 (2015): 405-423.
> [2] Rentzsch, Philipp, et al. "CADD-Splice—improving genome-wide variant effect prediction using deep learning-derived splice scores." Genome Medicine 13 (2021): 1-12.

---

### Official Review · Reviewer_zXNj · 2024-11-04

**Soundness:** 3
**Presentation:** 3
**Contribution:** 2
**Rating:** 6
**Confidence:** 4

**Summary:**

This manuscript presents a comprehensive benchmarking analysis for a wide range of DNA models in the task of predicting causal non-coding variants. The benchmark spans multiple model types, including supervised sequence-to-expression models, self-supervised DNA models, and baselines leveraging genome annotation-based summary statistics. The authors evaluate models on both Mendelian and complex traits, with a further distinction between disease and non-disease traits.

**Strengths:**

- This paper offers rigorous benchmarking for DNA models on causal variant prediction, ensuring a balanced comparison by controlling for factors such as distance to the transcription start site (TSS) and minor allele frequency (MAF) in the positive-negative matching.
- It evaluates an extensive array of model types, from supervised and self-supervised DNA models to non-ML baselines using CADD features, which adds robustness to the benchmark.

**Weaknesses:**

- The benchmark could be better positioned within the landscape of existing DNA benchmarks. While the authors have added a paragraph since my last review they haven't answered the question of what this particular benchmark brings in terms of insights on top of other existing ones. The discussion is just focused on the source of the variants. I appreciate that the CADD models provide a good baseline, and other benchmarks may not have this.

**Questions:**

- Please highlight any distinct model insights provided by this benchmark on top of other similar benchmark which is not based on the source of variants

---

> ### Author Response · Authors · 2024-12-02
>
> Thank you for your helpful comments and suggestions.  We have uploaded a revised version of our manuscript.
>
> > Feature extraction across model classes could be enhanced. For example, incorporating the L2 distance and cosine distance between reference and alternate embeddings (rather than only output features) in both supervised and self-supervised models could provide additional insights into model performance.
>
> > Please expand the feature set for supervised and self-supervised models to include L2 distances or cosine similarity metrics for embeddings
>
> We now compare different zero-shot scoring approaches for gLMs. Please see our updated text:
>
> “Good results have also been obtained comparing the embeddings of the alternate and reference alleles (DallaTorre et al., 2023; Mendoza-Revilla et al., 2024). We evaluate these different scoring approaches for each model (Table A.8) and choose the best performing one when benchmarking against other models (Table A.9).”
>
> > The benchmark would benefit from including variant classes that directly impact gene expression.
>
> > Please integrate gene expression variants from fine-mapped GTEx data (SuSiE) to enhance the benchmark’s relevance for expression-related causal variants
>
> The focus of our paper is on variants affecting high-level traits, which are known to have limited overlap with eQTLs (Mostafavi et al., 2023). We nevertheless have analyzed the overlap between complex trait causal variants and fine-mapped eQTL variants, and included our findings in the revised manuscript. Please see our updated text:
>
> “**eQTL colocalization**. We found that 103 putative causal variants for complex traits (9%) overlap with fine-mapped GTEx eQTL variants (Lonsdale et al., 2013; Wang et al., 2021); we found no such overlap for Mendelian trait variants, as expected given their low allele frequencies. The low overlap of complex trait and eQTL variants is well known and Mostafavi et al. (2023) discuss several hypotheses for the cause. We found that eQTL-overlapping variants are much easier to predict than non-eQTL-overlapping variants, across all model types (Figure 8D). We also note that Borzoi achieves a wide margin compared to other models and little is gained from ensembling. We observed that eQTL-overlapping variants are enriched in exonic variants (Fisher’s exact p = 8 × 10^−8) and, among non-exonic variants, they have lower TSS distances (Mann Whitney p = 4 × 10^−4), all of which could explain their increased predictability.”

---

> > ### Author Response · Authors · 2024-12-02
> >
> > > The benchmark could be better positioned within the landscape of existing DNA benchmarks. For instance, the pre-existing BEND benchmark covers causal variant prediction and includes ClinVar variants. This study’s use of OMIM and UKB variants, and the careful matching of positives and negatives, could add value if it demonstrates novel findings or highlights distinct model strengths. Otherwise it makes it hard for the field to decide which benchmark to use. Additionally, the authors should address recent benchmarks that have assessed DNA models on similar tasks using MPRA data https://www.biorxiv.org/content/10.1101/2024.02.29.582810v2.full.pdf and came to similar conclusions about the strength of supervised models over self-supervised.
> >
> > > Please provide a comparison between this benchmark and existing ones, such as BEND https://openreview.net/pdf?id=uKB4cFNQFg and https://www.biorxiv.org/content/10.1101/2024.02.29.582810v1 , highlighting any distinct model insights provided by the benchmark and some discussion about the difference in performance outcomes across model types
> >
> > We have expanded our discussion of related work to include Tang et al. (2024) and also calculated variant consequences in ClinVar to show its lack of coverage of the regulatory genome. Please see our updated text:
> >
> > “Tang et al. (2024) benchmark the ability of functional-genomics-supervised and self-supervised
> > models to predict non-coding variant effects on gene expression, but they cover neither Mendelian nor complex traits. BEND (Marin et al., 2024) and GV-Rep (Li et al., 2024) evaluate self-supervised models for the prediction of disease-associated variants from ClinVar (Landrum et al., 2020). While not documented, it is likely that these variants mostly cover Mendelian rather than complex diseases. Furthermore, expert-reviewed pathogenic variants in ClinVar are highly skewed towards coding and splice region variants, containing only a single promoter variant and no intergenic variants (Table A.7). Neither of these benchmarks establishes adequate baselines for this task. BEND includes a single early-generation functional-genomics-supervised model (Zhou & Troyanskaya, 2015), but does not include any conservation-based model, which are usually strong for this task (Benegas et al., 2023a). GV-Rep does not include any baseline.
> >
> > Thus, TraitGym is the only benchmark of causal non-coding variant prediction for both Mendelian and complex human traits. Furthermore, it is the only available framework to evaluate both the latest functional-genomics-supervised and self-supervised models, as well as strong non-neural baselines.”
> >
> > References:
> > Mostafavi, Hakhamanesh, et al. "Systematic differences in discovery of genetic effects on gene expression and complex traits." Nature Genetics 55.11 (2023): 1866-1875.

---

### Author Response · Authors · 2024-12-02
**General Response**

We thank the reviewers for their time and thoughtful feedback. We have uploaded a new version of our manuscript to address their questions and concerns.

We reiterate our main goals, which we believe will help advance future research at the intersection of machine learning and human genetics:
- A systematic benchmark for the challenging task of predicting causal variants for human Mendelian and complex traits, focusing on the often neglected non-coding part of the genome.
- An evaluation of the latest generation of broad classes of models: functional-genomics-supervised, self-supervised, and integrative.

In response to the reviewers’ questions and feedback, we have made the following modifications:
- **eQTLs** (Reviewers zXNj, Bb5Y): We analyze variants overlapping fine-mapped GTEx eQTLs (Fig. 8D).
- **Expanded negative sets** (Reviewers Bb5Y, 3FJZ): We provide and analyze additional expanded datasets with millions of negative variants (Fig. A.7).
- **Embedding metrics** (Reviewer zXNj): We evaluate additional gLM zero-shot scores such as L2 distance, cosine distance (Table A.8).
- **gLM-Promoter** (Reviewers Ub7o, Bb5Y): We have expanded our description of gLM-Promoter (Section A.2.2) and analyzed its performance across variant consequences (Figure A.8). gLM-Promoter was updated with the results of a longer training run and a slightly different architecture (Table A.6).
- **Related work** (Reviewers zXNj, Ub7o): We have clarified the differences with benchmarks in Tang and Koo (2024), BEND and ClinVar.
- **Additional experiments** (Reviewer Bb5Y): We evaluated the impact of different MAF thresholds (Fig. A.1), PIP thresholds (Fig. A.4), overlap with CADD variants (Fig. A.3) and matching by closest gene (Fig. A.2, A.5).
- **Additional metrics** (Reviewer 3FJZ): We report AUROC (Fig. A.6).
- **Pleiotropy**: We analyzed performance for variants affecting multiple traits (Fig. 8C).
- **Sei**: We evaluated Sei, a functional-genomics-supervised model from a different research group than Enformer and Borzoi.

We thank the reviewers for their feedback and would be happy to answer any further questions.

---

### Meta-Review · Area_Chair_cLQ9 · 2024-12-18

**Metareview:**

The paper presents TraitGym, a curated dataset designed to benchmark DNA sequence models for predicting causal genetic variants. The paper benchmarks various models and collects publicly available data sources for benchmarking.

Strengths included the comprehensiveness of the models and data and the colocation of benchmarking data within the framework for the community.

Several weaknesses were identified including alignment with existing threshold standards in the genetics community and organization and clarity of the paper.

The paper seems to be specifically about non-coding variants and does not address variant effect prediction for coding variants (per author response).  The abstract states as motivation, "...the field currently lacks consistently curated datasets with accurate labels, especially for non-coding variants, that are necessary to comprehensively benchmark these models...", but then states "In this work, we present TraitGym, a curated dataset of genetic variants that are either known to be causal or are strong candidates..." The motivating statement alludes to the focus on non-coding variants, but the main claim of the paper drops the distinction. Further, the title does not clarify that the focus is on non-coding variants. While the reviewers generally appreciated the distinction, there was some confusion. It is critical to align the claims of the paper with the evidence provided so that readers can properly place the paper among the literature. While the authors have offered many edits to the paper in the rebuttal period, which certainly improve on some of the weaknesses, the paper may need further edits to align the claims and evidence before acceptance.

**Additional Comments On Reviewer Discussion:**

The reviewers engaged in an extensive dialogue with the authors. In the discussion period the authors offered numerous edits to their paper and changes that certainly improve the paper. In particular, the authors pointed out some important papers known to the community that will help place this work in proper context.

---

### Decision · Program_Chairs · 2025-01-22

Reject